# ROBUSTNESS FOR FREE: ADVERSARIALLY ROBUST ANOMALY DETECTION THROUGH DIFFUSION MODEL

## ABSTRACT

Deep learning-based anomaly detection models have achieved remarkably high accuracy on commonly used benchmark datasets. However, the robustness of those models may not be satisfactory due to the existence of adversarial examples, which pose significant threats to the practical deployment of deep anomaly detectors. To tackle this issue, we propose an adversarially robust anomaly detector based on the diffusion model. There are two things that make diffusion models a perfect match for our task: 1) the diffusion model itself is a reconstruction-based modeling method whose reconstruction error can serve as a natural indicator of the anomaly score; 2) previous studies have shown that diffusion models can help purify the data for better adversarial robustness. In this work, we highlight that our diffusion model based method gains the adversarial robustness for free: the diffusion model will act both as an anomaly detector and an adversarial defender, thus no extra adversarial training or data purification is needed as in standard robust image classification tasks. We also extend our proposed method for certified robustness to $l_2$ norm bounded perturbations. Through extensive experiments, we show that our proposed method exhibits outstanding (certified) adversarial robustness while also maintaining equally strong anomaly detection performance on par with the state-of-the-art anomaly detectors on benchmark datasets.

## 1 INTRODUCTION

Anomaly detection aims at identifying data instances that are inconsistent with the majority of data, which has been widely applied in various domains such as industrial defect detection (Bergmann et al., 2019), IT infrastructure management (Sun et al., 2021), medical diagnostics (Fernando et al., 2021), and cyber security (Feng & Tian, 2021). Recently, deep learning (DL) based anomaly detection methods have achieved remarkable improvement over traditional anomaly detection strategies (Ruff et al., 2021; Pang et al., 2021). DL-based methods take the advantage of neural networks to estimate the *anomaly score* of a data instance which reflects how likely it is an anomaly. One common practice defines anomaly score as the *reconstruction error* between the original data instance and the recovered one decoded by a symmetric neural network model (e.g., autoencoder) (Hawkins et al., 2002; Chen et al., 2017). The insight that the reconstruction error can serve as anomaly score is that the model trained on normal data usually cannot reproduce anomalous instances (Bergmann et al., 2021), thus a high reconstruction error for a data instance indicates a larger probability of it being an anomaly.

Though DL-based anomaly detection methods have achieved remarkably high accuracy on commonly used benchmark datasets (Yu et al., 2021; Lee et al., 2022a), the robustness of the detection models is still unsatisfactory due to the existence of adversarial examples (Goodge et al., 2020; Lo et al., 2022), which poses significant threats to the practical deployment of deep anomaly detectors. Specifically, an imperceptible perturbation on the input data could cause a well-trained anomaly detector to return incorrect detection results. Figure 1 shows a simple case of how such an adversarial attack can disrupt OCR-GAN (Liang et al., 2022) which is a recent deep image anomaly detector. We observe that an anomalous "hazelnut" in the upper row, when added with an invisible noise, could cheat the detector to output a low anomaly score; while the normal "hazelnut" in the lower row can also be perturbed to make the detector raise a false alarm with a high anomaly score. In fact, such a robustness issue is not unique to OCR-GAN, but a common problem for various state-of-the-art deep anomaly detection models (as will be seen in our later experiments in Section 3).

To tackle this issue, we explore the possibility of using the *diffusion model* to achieve adversarially robust anomaly detection. As a powerful class of generative models, diffusion models (Ho et al., 2020; Nichol & Dhariwal, 2021) are capable of generating samples with high quality, beating GANs in image synthesis (Dhariwal & Nichol, 2021). Specifically, diffusion models first construct a diffusion process to convert the data into standard Gaussian noise by gradually adding random noise, and then learn the generative process to reverse the diffusion process and generate samples from the noise by denoising one step at a time. There are two aspects about diffusion models that make them a perfect match for building an adversarially robust anomaly detector: 1) *anomaly detection capability*, as the diffusion model itself is a reconstruction-based modeling method whose reconstruction error can serve as a natural indicator of the anomaly score. A diffusion model trained on normal data ideally can reconstruct anomalies as normal ones through the diffusion and reverse generative process, thus bringing high reconstruction scores for anomalies compared with normal instances; 2) *adversarial robustness*, as previous studies have shown that diffusion models can be used as a data purifier to mitigate adversarial noises for better robustness (Nie et al., 2022) in supervised learning tasks, which suggests its potential in defending adversarial examples in the anomaly detection task.

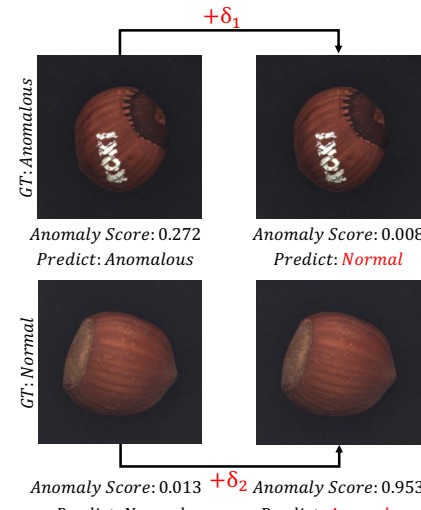

Figure 1: An adversarial example on OCR-GAN. $\delta$ refers to invisible perturbations. "GT" denotes "Ground Truth".

Based on the nice properties of diffusion models, we propose a novel adversarially robust anomaly detection method, inside which the diffusion model acts both as an *anomaly detector* and an *adversarial defender*. The introduction of the diffusion model enables us to gain adversarial robustness *for free*, as no extra adversarial training or data purification is needed. Note that our design is fundamentally different from the purification-based adversarial robust models in standard image classification tasks (Nie et al., 2022) where an extra external purifier (e.g., diffusion model) is needed before the actual classifier for robust classification, which is not needed in our design[1].

We summarize our contributions as follows:

- We build a unified adversarial attack framework for various kinds of anomaly detectors to facilitate the adversarial robustness study in the anomaly detection domain, through which we systematically evaluate the adversarial robustness of state-of-the-art deep anomaly detection models.

- We propose an anomaly detection method based on the diffusion model, which gains adversarial robustness for free: the diffusion model acts both as an anomaly detector and an adversarial defender, without extra need for adversarial training or data purification as in standard robust image classification tasks. We also extend our method for certified robustness to $l_2$ norm perturbations through randomized smoothing which provides additional robustness guarantees.

- We conduct extensive experiments and show that our method exhibits outstanding (certified) adversarial robustness, while also maintaining equally strong anomaly detection performance on par with the state-of-the-art anomaly detectors on benchmark datasets (Bergmann et al., 2019).

## 2  RELATED WORK

**Anomaly Detection Methods.** Existing anomaly detection methods can be roughly categorized into two kinds: *reconstruction-based* and *feature-based*. One commonly used *reconstruction-based* approach for anomaly detection is to train the autoencoder and use the $l_p$ norm distance between input and its reconstruction as the anomaly score (Hawkins et al., 2002; Chen et al., 2017; Zhou & Paffenroth, 2017). Bergmann et al. (2018) replace $l_p$ distance with SSIM (Wang et al., 2004) to

---

[1]In fact, the strategy of using the diffusion model as a purifier before another anomaly detector will not work, as the purifier will break the anomaly signals.

have a better measure for perceptual similarity. Another more advanced branch of reconstruction-based models combines autoencoder with GAN, where the generator of the GAN is implemented using autoencoder (Hou et al., 2021; Liang et al., 2022; Akçay et al., 2019). These methods additionally incorporate the anomaly score with the similarity between the features of the input and the reconstructed images extracted from the discriminator to boost performance on categories that are difficult to reconstruct accurately. *Feature-based* methods use pre-trained Resnet and vision transformer (Yu et al., 2021), or pre-trained neural networks with feature adaptation (Lee et al., 2022a) to extract discriminative features for normal images, and estimate distribution of these normal features by Flow-based model (Gudovskiy et al., 2022; Rudolph et al., 2022), KNN (Reiss et al., 2021), or Gaussian distribution modeling (Li et al., 2021). These methods calculate the anomaly score using the distance from the features of test images to the established distribution for features of normal images.

**Adversarial Attacks and Defenses for Anomaly Detectors.** To the best of our knowledge, existing attack and defense strategies for anomaly detectors only focus on autoencoder-based models. Goodge et al. (2020) consider perturbations to anomalous data that make the model to categorize them as the normal class by reducing reconstruction error. For defense, they propose APAE using approximate projection and feature weighting to improve adversarial robustness. Lo et al. (2022) extend the similar attack strategy to both normal and anomalous data and propose Principal Latent Space as a defense strategy to perform adversarially robust novelty detection (i.e., only semantic shift anomalies are considered). While they achieve a certain level of robustness, their performances on clean anomaly detection tasks are yet far from satisfactory.

**Diffusion Models.** As a class of powerful generative models, diffusion models have attracted the most recent attention due to their high sample quality and strong mode coverage (Sohl-Dickstein et al., 2015; Ho et al., 2020; Nichol & Dhariwal, 2021). Recently, Nie et al. (2022) used diffusion models to purify adversarial perturbations for downstream robust classification, and present empirically strong robustness. Wolleb et al. (2022) adopt deterministic DDIM (Song et al., 2020) for supervised anomaly localization. Wyatt et al. (2022) solve the same task under an unsupervised scenario using DDPM (Ho et al., 2020) with partial diffusion strategy and simplex noise. Note that they are pixel-level anomaly detection methods which are not directly comparable to our image-level anomaly detection. Moreover, diffusion models have not been studied to improve the adversarial robustness of anomaly detectors.

## 3 BUILDING ADVERSARIAL ATTACKS FOR ANOMALY DETECTORS THROUGH A UNIFIED FRAMEWORK

To facilitate the adversarial robustness study on various kinds of anomaly detectors, we first build a unified adversarial attack framework in the context of anomaly detection. We consider the adversarial perturbations to be imperceptible, i.e, their existence will not flip the ground truth class of the image (label-preserving). The general goal of the unified attack framework is to make detectors return incorrect detection results by reducing anomaly scores for anomalous samples and increasing anomaly scores for normal samples. In particular, we take commonly used Projected Gradient Descent (PGD) attack (Madry et al., 2018) as an example to illustrate our attack formulation.

**PGD Attack on Anomaly Detector.** Consider a sample $\mathbf{x} \in \mathbb{R}^d$ from the test dataset with label $\mathrm{y} \in \{-1, 1\}$ (where "$-1$" denotes the anomalous class and "1" indicates the normal class), and a well-trained anomaly detector $A_{\boldsymbol{\theta}} : \mathbb{R}^d \to \mathbb{R}$ that computes an anomaly score for each data sample. We define the optimization objective of PGD attack on the anomaly detector as: $\arg\max_{\mathbf{x}} L_{\boldsymbol{\theta}}(\mathbf{x}, \mathrm{y}) = \mathrm{y} A_{\boldsymbol{\theta}}(\mathbf{x})$, where y guides the direction of perturbing $\mathbf{x}$ to increase or decrease its anomaly score. Depending on the perturbation constraint, adversarial examples can be generated by $l_\infty$-norm or $l_2$-norm bounded PGD, respectively as:

$$\mathbf{x}_{n+1} = P^{l_\infty}_{\mathbf{x},\epsilon}\{\mathbf{x}_n + \alpha \cdot \mathrm{sgn}(\nabla_{\mathbf{x}_n} L_{\boldsymbol{\theta}}(\mathbf{x}_n, \mathrm{y})\} \tag{3.1}$$

$$\mathbf{x}_{n+1} = P^{l_2}_{\mathbf{x},\epsilon}\left\{\mathbf{x}_n + \alpha \frac{\nabla_{\mathbf{x}_n} L_{\boldsymbol{\theta}}(\mathbf{x}_n, \mathrm{y})}{\|\nabla_{\mathbf{x}_n} L_{\boldsymbol{\theta}}(\mathbf{x}_n, \mathrm{y})\|}\right\} \tag{3.2}$$

where $\alpha$ is the step size, $n \in [0, N-1]$ is the current step of in total $N$ iterations, and $\mathbf{x}_0 = \mathbf{x}$. $P^{l_p}_{\mathbf{x},\epsilon}\{\cdot\}$ denotes the projection on $\mathbf{x}_{n+1}$ such that $\|\mathbf{x}_{n+1} - \mathbf{x}\|_p \le \epsilon$. The final adversarial example

is generated by $\mathbf{x}_{adv} = \mathbf{x}_N$. This attacking strategy encapsulates previous works on adversarial examples for anomaly detectors, where only autoencoder-based models were considered (Lo et al., 2022; Goodge et al., 2020). The anomaly score can be specified as $A_{\boldsymbol{\theta}}(\mathbf{x}) = \|D(E(\mathbf{x})) - \mathbf{x}\|$ to accommodate to their scenarios, where $D$ denotes the decoder and $E$ corresponds to the encoder.

**Robustness Evaluation on Existing Anomaly Detectors.** Based on the unified PGD attack, we systematically evaluate the adversarial robustness of the state-of-the-art detectors with various model architectures. Table 1 demonstrates the efficacy of the attack in disclosing the vulnerability of existing anomaly detectors: the AUC scores of these advanced anomaly detectors drop to as low as $0\%$ under adversarial perturbations with $l_{\infty}$ norm less than $2/255$ on *Toothbrush* dataset from benchmark MVTec AD (Bergmann et al., 2019). This suggests that current anomaly detectors suffer from fragile robustness on adversarial data, which urges us to build adversarially robust anomaly detectors that can achieve excellent detection performance and strong adversarial robustness simultaneously.

Table 1: Standard AUC and robust AUC against $l_{\infty}$-PGD ($\epsilon = 2/255$) attacks on *Toothbrush* dataset from benchmark MVTec AD, obtained by various anomaly detection SOTAs.

| Method | Standard AUC | Robust AUC |
|---|---|---|
| OCR-GAN (Liang et al., 2022) | 96.7 | 0 |
| SPADE (Cohen & Hoshen, 2020) | 88.9 | 0 |
| CFlow (Gudovskiy et al., 2022) | 85.3 | 0 |
| FastFlow (Yu et al., 2021) | 94.7 | 0 |
| CFA (Lee et al., 2022a) | 100 | 0 |

## 4 ADVERSARIALLY ROBUST ANOMALY DETECTION

Before we introduce our diffusion-based robust anomaly detection method, we first give a brief review on diffusion models (Sohl-Dickstein et al., 2015; Ho et al., 2020; Nichol & Dhariwal, 2021).

### 4.1 PRELIMINARIES ON DIFFUSION MODELS

DDPM (Ho et al., 2020) defines a $T$ steps diffusion process $q(\mathbf{x}_{1:T}|\mathbf{x}_0) := \prod_{t=1}^{T} q(\mathbf{x}_t|\mathbf{x}_{t-1})$ parameterized by a well behaved variance schedule $\beta_1, \ldots, \beta_T$ as $q(\mathbf{x}_t|\mathbf{x}_{t-1}) := \mathcal{N}(\mathbf{x}_t; \mathbf{x}_{t-1}\sqrt{1 - \beta_t}, \beta_t I)$, which iteratively transforms an unknown data distribution $q(\mathbf{x}_0)$ to standard Gaussian $q(\mathbf{x}_T) = \mathcal{N}(0, \mathbf{I})$. The generative process $p_{\boldsymbol{\theta}}(\mathbf{x}_{0:T}) := p(\mathbf{x}_T)\prod_{t=1}^{T} p_{\boldsymbol{\theta}}(\mathbf{x}_{t-1}|\mathbf{x}_t)$ is learned to approximate each $q(\mathbf{x}_{t-1}|\mathbf{x}_t)$ using neural networks as follows:

$$p_{\boldsymbol{\theta}}(\mathbf{x}_{t-1}|\mathbf{x}_t) := \mathcal{N}(\mathbf{x}_{t-1}; \boldsymbol{\mu}_{\boldsymbol{\theta}}(\mathbf{x}_t, t), \boldsymbol{\Sigma}_{\boldsymbol{\theta}}(\mathbf{x}_t, t)) \tag{4.1}$$

A noticeable property of the diffusion process is that it allows directly sampling $\mathbf{x}_t$ at an arbitrary timestep $t$ given $\mathbf{x}_0$. Using the notation $\alpha_t := 1 - \beta_t$ and $\overline{\alpha_t} := \prod_{s=1}^{t} \alpha_s$, we have

$$\mathbf{x}_t = \sqrt{\overline{\alpha_t}}\mathbf{x}_0 + \sqrt{1 - \overline{\alpha_t}}\boldsymbol{\epsilon}, \quad \boldsymbol{\epsilon} \in \mathcal{N}(0, \mathbf{I}) \tag{4.2}$$

This property makes it possible to quickly sample $\mathbf{x}_t$. For training the diffusion model, motivated by the connection to generative score matching (Song & Ermon, 2019; 2020), Ho et al. (2020) show that directly predicting the noise term $\boldsymbol{\epsilon}$ results in higher sample quality, especially when combined with a simplified objective without learning signals for $\boldsymbol{\Sigma}_{\boldsymbol{\theta}}(\mathbf{x}_t, t)$:

$$L_{\text{simple}} = \mathbb{E}_{t,\mathbf{x}_0,\boldsymbol{\epsilon}}[\|\boldsymbol{\epsilon} - \boldsymbol{\epsilon}_{\boldsymbol{\theta}}(\mathbf{x}_t, t)\|]. \tag{4.3}$$

In this paper, we follow Nichol & Dhariwal (2021) and train the diffusion model using a hybrid loss for better sample quality with fewer generation steps. More details can be found in Appendix A.

### 4.2 FREERAD: ADVERSARIALLY ROBUST ANOMALY DETECTION FOR FREE

Based on the diffusion model, we now introduce our proposed **R**obust **A**nomaly **D**etection for **Free** method, termed as **FreeRAD**. FreeRAD consists of two parts: *robust reconstruction*, which aims to reconstruct the normal input in a robust manner, and *anomaly score calculation*, which aims to calculate the final anomaly score based on the robust reconstruction error.

**Robust Reconstruction:** Robust reconstruction is the first step for our FreeRAD method and is the key to achieving adversarially robust anomaly detection. Since the diffusion model training procedure is essentially predicting noise added in the diffusion process and then denoising, its reconstruction error can serve as a natural indicator of the anomaly score. Specifically, as shown in Figure 2, for normal data, the reconstruction is nearly identical to the input. For anomaly data, the diffusion model (after adding noise and denoising) could "repair" the anomaly regions, thus obtaining high reconstruction error, which could be easily detected as anomalies. Now let's consider adversarial robustness in anomaly detection. Note that one basic assumption of adversarial examples is that the perturbation is usually imperceivable, e.g., with small $L_p$ norms. In the diffusion process, if we add sufficiently large Gaussian noise to the input data, such adversarial perturbations would be dominated by the added Gaussian and thus be invalid. After the reverse diffusion (denoising) process, the reconstruction could still recover it to normal and thus obtain a high reconstruction error as shown in Figure 2. This suggests that FreeRAD is indeed robust to adversarial data perturbations.

Algorithm 1 summarizes the main steps for robust reconstruction. Specifically, to perform adversarially robust reconstruction, we first choose the diffusion steps $k$ and apply Eq. 4.2 on $\mathbf{x}$ to obtain diffused images $\mathbf{x}_k$. Unlike the diffusion model training process, here we do not need to diffuse the data into complete Gaussian noise (a large $k$). Instead, we pick a moderate number of $k$ for noise injection and start denoising thereafter, similar to Nie et al. (2022). Note that $k$ should be chosen such that the amount of Gaussian noise is dominating the adversarial perturbations and anomaly signals while the high-level features of the input data are still preserved for reconstruction. In terms of the denoising process, a typical full-shot setting uses the full $k$ denoising steps: in each step $t$, we iteratively predict the true input $\mathbf{x}$ given the current diffused data $\mathbf{x}_t$, termed $\widetilde{\mathbf{x}}_0$, then sampling the new iterate $\mathbf{x}_{t-1}$ according to the current prediction $\widetilde{\mathbf{x}}_0$ and the current diffused data $\mathbf{x}_t$.

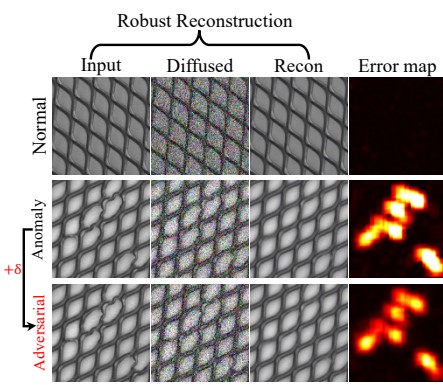

Figure 2: Reconstruction results of normal data, anomalous data, and adversarially perturbed data using our model. The observed reconstruction is robust to adversarial noise.

---

**Algorithm 1** Full-shot Robust Reconstruction in FreeRAD

---

**Input:** Test images: $\mathbf{x}$, diffusion steps: $k(k \leq T)$
**Output:** Reconstructions of $\mathbf{x}$: $\widetilde{\mathbf{x}}$

1: $\mathbf{x}_0 = \mathbf{x}$
2: $\boldsymbol{\epsilon} \sim \mathcal{N}(0, \mathbf{I})$
3: $\mathbf{x}_k = \sqrt{\overline{\alpha}_k}\mathbf{x}_0 + \sqrt{1 - \overline{\alpha}_k}\boldsymbol{\epsilon}$
4: **for** $t = k, \ldots, 1$ **do**                                            ▷ full-shot denoising
5:     $\widetilde{\mathbf{x}}_0 = \frac{1}{\sqrt{\overline{\alpha}_t}}(\mathbf{x}_t - \sqrt{1 - \overline{\alpha}_t}\boldsymbol{\epsilon}_{\boldsymbol{\theta}}(\mathbf{x}_t, t))$
6:     **if** $t > 1$ **then**
7:         $\mathbf{z} \sim \mathcal{N}(0, \mathbf{I})$
8:         $\mathbf{x}_{t-1} = \frac{\sqrt{\overline{\alpha}_{t-1}}\beta_t}{1 - \overline{\alpha}_t}\widetilde{\mathbf{x}}_0 + \frac{\sqrt{\overline{\alpha}_t}(1 - \overline{\alpha}_{t-1})}{1 - \overline{\alpha}_t}\mathbf{x}_t + \sqrt{\boldsymbol{\Sigma}_{\boldsymbol{\theta}}(\mathbf{x}_t, t)}\mathbf{z}$
9:     **end if**
10: **end for**
11: $\widetilde{\mathbf{x}} = \widetilde{\mathbf{x}}_0$

---

**Anomaly Score Calculation:** To calculate the final anomaly score in a robust and stable manner, we first calculate the Multiscale Reconstruction Error Map (denoted as $\mathrm{Err}_{\mathrm{ms}}$), which considers both pixel-wise and patch-wise reconstruction errors. Specifically, for each scale $l$ in $L = \{1, \frac{1}{2}, \frac{1}{4}, \frac{1}{8}\}$, we first calculate the error map $\mathrm{Err}(\mathbf{x}, \widetilde{\mathbf{x}})_l$ between the downsampled input $\mathbf{x}^l$ and the downsampled reconstruction $\widetilde{\mathbf{x}}^l$ with $\frac{1}{C}\sum_{c=1}^{C}(\mathbf{x}^l - \widetilde{\mathbf{x}}^l)^2_{[c,:,:]}$ where the square operator is abused here for element-wise square operation, then unsampled to the original resolution. The final $\mathrm{Err}_{\mathrm{ms}}$ is obtained by averaging each scale's error map and applying a mean filter for better stability similar to Zavrtanik et al. (2021): $\mathrm{Err}_{\mathrm{ms}}(\mathbf{x}, \widetilde{\mathbf{x}}) = (\frac{1}{N_L}\sum_{l \in L}\mathrm{Err}(\mathbf{x}, \widetilde{\mathbf{x}})_l) * f_{s \times s}$ where $f_{s \times s}$ is the mean filter of size $s \times s$,

$*$ is the convolution operation. Similar to Pirnay & Chai (2022), we take the pixel-wise maximum of the absolute deviation of the $\mathrm{Err}_{\mathrm{ms}}(\mathbf{x}, \widetilde{\mathbf{x}})$ on normal training data as the scalar anomaly score. Due to space limits, we leave the complete anomaly score calculation algorithm in Appendix B.2.

**One-shot Denoising:** One major problem with full-shot denoising (Algorithm 1) is that the denoising procedure is time consuming, making it unacceptable for real-time anomaly detection in critical situations (Sun et al., 2021). Moreover, extra reconstruction error can also be introduced due to the multiple sampling steps in the full-shot denoising process. To overcome these challenges, we investigate the arbitrary-shot denoising process allowing fewer denoising steps, with the details shown in Appendix B.1. Based on our results (see Appendix D.2) we observe that one-shot denoising (reducing the for loop in Line 4 of Algorithm 1 into one iteration) is sufficient to produce an accurate reconstruction result with $\mathcal{O}(1)$ inference-time efficiency. Under such cases, the robust reconstruction in FreeRAD reduces to the simple 3-step version shown in Algorithm 2. Such a one-shot idea has also been adopted in Carlini et al. (2022) for robust image classification. By default, we use one-shot robust reconstruction for all experiments in Section 5.

---

**Algorithm 2** One-shot Robust Reconstruction in FreeRAD

---

    **Input:** Test images: $\mathbf{x}$, diffusion step: $k(k \leq T)$
    **Output:** Reconstructions of $\mathbf{x}$: $\widetilde{\mathbf{x}}$
1: $\boldsymbol{\epsilon} \sim \mathcal{N}(0, \mathbf{I})$
2: $\mathbf{x}_k = \sqrt{\overline{\alpha}_k}\mathbf{x} + \sqrt{1 - \overline{\alpha}_k}\boldsymbol{\epsilon}$
3: $\widetilde{\mathbf{x}} = \frac{1}{\sqrt{\overline{\alpha}_k}}(\mathbf{x}_k - \sqrt{1 - \overline{\alpha}_k}\boldsymbol{\epsilon}_{\boldsymbol{\theta}}(\mathbf{x}_k, k))$            $\triangleright$ one-shot denoising process

---

## 5 EXPERIMENTS

We compare our proposed FreeRAD with five state-of-the-art anomaly detectors on both clean input and adversarially perturbed input. FreeRAD shows a competitive robustness performance compared with defense-enabled anomaly detector baselines, and maintains robust even under stronger adaptive attacks. Finally, we further extend FreeRAD for certified robustness to $l_2$ norm perturbations.

### 5.1 EXPERIMENTAL SETTINGS

**Dataset and Model Implementation.** We perform experiments on widely used MVTec Anomaly Detection benchmark (Bergmann et al., 2019). MVTec AD comprises 15 sub-datasets with a total of 5354 high-resolution images from the real world. Among these sub-datasets, the category for 10 of them are about specific objects (e.g., toothbrush, transistor, hazelnut), and the other 5 sub-datasets are about specific textures (e.g., leather, wood). We resize all images to 256×256 resolution in our experiments. We implement the diffusion model based on Nichol & Dhariwal (2021) using U-Net backbone (Ronneberger et al., 2015). We set the total iteration step as $T = 1000$ for all experiments. During inference stage, we choose the diffusion step $k \in \{50, 100, 200, 300\}$ for different categories (see Appendix D.1 for sensitivity test). More hyperparameters are described in Appendix C.1.

**Adversarial Attacks.** We adopt commonly used PGD attack (Madry et al., 2018) to compare with the state-of-the-art anomaly detection models and defense-enabled anomaly detectors. Additionally, we also consider the BPDA and EOT attack (Athalye et al., 2018a) for better robustness evaluations on defense-enabled anomaly detectors. We set the attack strength $\epsilon = 2/255$ for $l_\infty$-norm attacks and $\epsilon = 0.2$ for $l_2$-norm attacks to ensure imperceptible attack perturbations.

**Evaluation Metric.** We use the widely-adopted AUC (area under the receiver operating characteristic curve) to evaluate the performance of anomaly detection. Specifically, we consider *standard AUC* and *robust AUC*. The standard AUC evaluates the performance on the clean test data, while the robust AUC evaluates the performance on the adversarially perturbed test examples.

### 5.2 COMPARISON WITH THE STATE-OF-THE-ART ANOMALY DETECTORS

We compare our method FreeRAD with five state-of-the-art methods for image anomaly detection: SPADE (Cohen & Hoshen, 2020), OCR-GAN (Liang et al., 2022), CFlow (Gudovskiy et al., 2022), FastFlow (Yu et al., 2021), and CFA (Lee et al., 2022a), against the $l_\infty$-PGD and $l_2$-PGD attacks.

Table 2: Standard AUC (in parenthesis) and robust AUC against $l_\infty$-PGD attacks ($\epsilon = 2/255$) on MVTec AD dataset, obtained by different state-of-the-art anomaly detectors and ours.

| | Category | OCR-GAN | SPADE | CFlow | FastFlow | CFA | FreeRAD |
|---|---|---|---|---|---|---|---|
| Texture | Carpet | $0^{(76.6)}$ | $0^{(92.8)}$ | $0^{(98.6)}$ | $0^{(\mathbf{99.7})}$ | $0^{(99.4)}$ | $\mathbf{70.5}^{(82.7)}$ |
| | Grid | $0^{(97)}$ | $0^{(47.3)}$ | $0^{(96.6)}$ | $0^{(\mathbf{100})}$ | $0^{(99.6)}$ | $\mathbf{99.8}^{(\mathbf{100})}$ |
| | Leather | $0^{(90.7)}$ | $0^{(95.4)}$ | $0^{(\mathbf{100})}$ | $6.6^{(\mathbf{100})}$ | $2.0^{(\mathbf{100})}$ | $\mathbf{97.8}^{(\mathbf{100})}$ |
| | Tile | $0^{(95.6)}$ | $0^{(96.5)}$ | $0^{(99.6)}$ | $1.3^{(\mathbf{100})}$ | $0.1^{(99.3)}$ | $\mathbf{93.9}^{(99.2)}$ |
| | Wood | $0^{(95.4)}$ | $0^{(95.8)}$ | $0^{(99.7)}$ | $0^{(\mathbf{99.9})}$ | $0^{(99.7)}$ | $\mathbf{95.2}^{(98.3)}$ |
| Object | Bottle | $0^{(97.7)}$ | $0^{(97.2)}$ | $0^{(\mathbf{100})}$ | $0^{(\mathbf{100})}$ | $0.1^{(\mathbf{100})}$ | $\mathbf{88.1}^{(\mathbf{100})}$ |
| | Cable | $0^{(71.5)}$ | $0^{84.8}$ | $0^{(98.7)}$ | $0^{(67.4)}$ | $0.8^{(\mathbf{99.8})}$ | $\mathbf{38.9}^{(79.5)}$ |
| | Capsule | $0^{(80.4)}$ | $0^{(89.7)}$ | $0^{(93.7)}$ | $8.9^{(\mathbf{99.2})}$ | $0^{(97)}$ | $\mathbf{53.5}^{(93.9)}$ |
| | Hazelnut | $0^{(97.7)}$ | $0^{(88.1)}$ | $0^{(99.9)}$ | $0^{(99.5)}$ | $0.1^{(\mathbf{100})}$ | $\mathbf{91.5}^{(97.5)}$ |
| | Metal Nut | $0^{(82.6)}$ | $0^{(71)}$ | $0^{(\mathbf{100})}$ | $0^{(98.2)}$ | $0^{(\mathbf{100})}$ | $\mathbf{85.9}^{(93.5)}$ |
| | Pill | $0^{(80.8)}$ | $0^{(80.1)}$ | $0^{(93.2)}$ | $0^{(97.8)}$ | $0^{(98)}$ | $\mathbf{39}^{(97.2)}$ |
| | Screw | $0^{(\mathbf{99.4})}$ | $0^{(66.7)}$ | $0^{(79)}$ | $6.6^{(91.1)}$ | $0^{(95.5)}$ | $\mathbf{87.6}^{(99.3})$ |
| | Toothbrush | $0^{(96.7)}$ | $0^{(88.9)}$ | $0^{(85.3)}$ | $0^{(94.7)}$ | $0^{(\mathbf{100})}$ | $\mathbf{95.8}^{(\mathbf{100})}$ |
| | Transistor | $0^{(75)}$ | $0^{(90.3)}$ | $0^{(98.3)}$ | $0^{(\mathbf{99.4})}$ | $0^{(\mathbf{100})}$ | $\mathbf{74.5}^{(93.7)}$ |
| | Zipper | $0^{(80.4)}$ | $0^{(96.6)}$ | $0^{(97.5)}$ | $17.5^{(99.6)}$ | $0^{(99.7)}$ | $\mathbf{96.2}^{(\mathbf{100})}$ |
| | **Average** | $0^{(87.8)}$ | $0^{(85.4)}$ | $0^{(96.0)}$ | $2.3^{(98.5)}$ | $0.2^{(\mathbf{99.2})}$ | $\mathbf{80.5}^{(95.7)}$ |

Table 3: Standard AUC (in parenthesis) and Robust AUC against $l_2$-PGD attacks ($\epsilon = 0.2$) on MVTec AD dataset, obtained by different state-of-the-art anomaly detectors and ours.

| | Category | OCR-GAN | SPADE | CFlow | FastFlow | CFA | FreeRAD |
|---|---|---|---|---|---|---|---|
| Texture | Carpet | $18.5^{(76.6)}$ | $27.1^{(92.8)}$ | $13.5^{(98.6)}$ | $18^{(\mathbf{99.7})}$ | $65.1^{(99.4)}$ | $\mathbf{76.6}^{(82.7)}$ |
| | Grid | $0^{(97)}$ | $4.1^{(47.3)}$ | $0^{(96.6)}$ | $0^{(100)}$ | $50^{(99.6)}$ | $\mathbf{99.9}^{(\mathbf{100})}$ |
| | Leather | $0^{(90.7)}$ | $16.5^{(95.4)}$ | $9.4^{(100)}$ | $35.4^{(100)}$ | $77.6^{(100)}$ | $\mathbf{99.9}^{(\mathbf{100})}$ |
| | Tile | $7.4^{(95.6)}$ | $45.9^{(96.5)}$ | $7.8^{(99.6)}$ | $30.5^{(100)}$ | $72.4^{(99.3)}$ | $\mathbf{93.1}^{(99.2)}$ |
| | Wood | $0^{(95.4)}$ | $11^{(95.8)}$ | $18.1^{(99.7)}$ | $22^{(\mathbf{99.9})}$ | $61.8^{(99.7)}$ | $\mathbf{95.5}^{(98.3)}$ |
| Object | Bottle | $0.1^{(97.7)}$ | $0^{(97.2)}$ | $48.5^{(100)}$ | $2.2^{(100)}$ | $74.6^{(100)}$ | $\mathbf{95.5}^{(\mathbf{100})}$ |
| | Cable | $3.2^{(71.5)}$ | $0.9^{84.8}$ | $19.2^{(98.7)}$ | $0.3^{(67.4)}$ | $\mathbf{69.5}^{(\mathbf{99.8})}$ | $65.7^{(79.5)}$ |
| | Capsule | $0^{(80.4)}$ | $0^{(89.7)}$ | $1.6^{(93.7)}$ | $13.8^{(\mathbf{99.2})}$ | $1.7^{(97)}$ | $\mathbf{68.1}^{(93.9)}$ |
| | Hazelnut | $18.5^{(97.7)}$ | $0^{(88.1)}$ | $4.9^{(99.9)}$ | $0.8^{(99.5)}$ | $47.2^{(100)}$ | $\mathbf{94.3}^{(97.5)}$ |
| | Metal Nut | $2.8^{(82.6)}$ | $0^{(71)}$ | $4.4^{(100)}$ | $1.7^{(98.2)}$ | $14.3^{(100)}$ | $\mathbf{87.9}^{(93.5)}$ |
| | Pill | $2.7^{(80.8)}$ | $0.4^{(80.1)}$ | $0^{(93.2)}$ | $0^{(97.8)}$ | $3.3^{(98)}$ | $\mathbf{80.3}^{(97.2)}$ |
| | Screw | $0^{(\mathbf{99.4})}$ | $0^{(66.7)}$ | $0^{(79)}$ | $6.6^{(91.1)}$ | $0^{(95.5)}$ | $\mathbf{91.8}^{(99.3})$ |
| | Toothbrush | $0^{(96.7)}$ | $0^{(88.9)}$ | $18.3^{(85.3)}$ | $3.6^{(94.7)}$ | $38.3^{(100)}$ | $\mathbf{99.4}^{(\mathbf{100})}$ |
| | Transistor | $1.7^{(75)}$ | $4.8^{(90.3)}$ | $8.8^{(98.3)}$ | $0.4^{(\mathbf{99.4})}$ | $53.7^{(100)}$ | $\mathbf{84.3}^{(93.7)}$ |
| | Zipper | $0^{(80.4)}$ | $3.2^{(96.6)}$ | $0^{(97.5)}$ | $19.3^{(99.6)}$ | $29.2^{(99.7)}$ | $\mathbf{99.2}^{(\mathbf{100})}$ |
| | **Average** | $3.7^{(87.8)}$ | $7.59^{(85.4)}$ | $10.3^{(96.0)}$ | $9.9^{(98.5)}$ | $43.9^{(\mathbf{99.2})}$ | $\mathbf{88.8}^{(95.7)}$ |

Table 2 presents the robustness performance against $l_\infty$-PGD attacks ($\epsilon = 2/255$) on MVTec AD dataset. Table 3 shows the robustness performance against $l_2$-PGD attacks ($\epsilon = 0.2$).

From Table 2 we observe that our method largely outperforms previous methods regarding robust AUC against $l_\infty$-PGD attacks ($\epsilon = 2/255$). Specifically, our method improves robust AUC on all 15 categories of MVTec AD and obtains the average robust AUC $80.5\%$ with the improvement of at least $78.2\%$. In Table 3, we can see that our method improves average robust AUC against $l_2$-PGD attacks ($\epsilon = 0.2$) by $44.9\%$ and achieves $88.8\%$ robust AUC. In the meantime, we can observe that in terms of anomaly detection performance on clean data, the average standard AUC obtained by our method is on par with the state-of-the-art methods such as CFlow (Gudovskiy et al., 2022), FastFlow (Yu et al., 2021), and CFA (Lee et al., 2022a), while beating OCR-GAN (Liang et al., 2022) and SPADE Cohen & Hoshen (2020). These results clearly demonstrate the effectiveness of our proposed method in defending against $l_\infty$-PGD and $l_2$-PGD attacks, while also maintaining strong anomaly detection performance on benchmark datasets.

### 5.3 COMPARISON WITH DEFENSE-ENABLED ANOMALY DETECTORS

In this section, We compare our method FreeRAD with APAE (Goodge et al., 2020) and PLS (Lo et al., 2022), two defense-enabled anomaly detection methods. We perform the same PGD attacks as in Section 5.2. Additionally, since APAE has an optimization loop in their defense process which is hard to backpropagate, we further adopt the BPDA attack (Athalye et al., 2018a) designed specifically for obfuscated gradient defenses to evaluate both our FreeRAD and APAE for a fair comparison. Table 4 shows the comparison between our FreeRAD method and defense-enabled baselines against PGD and BPDA attacks. We can clearly observe that FreeRAD outperforms them under all attacks, with a substantial improvement of $27.6\% \sim 58.3\%$ regarding the average robust AUC over all categories of MVTec AD. Furthermore, our method even largely improves the average standard AUC by $31.0\%$ on clean data compared with APAE and PLS.

Table 4: Average standard AUC and robust AUC against $l_\infty$-PGD/BPDA ($\epsilon = 2/255$), $l_2$-PGD/BPDA ($\epsilon = 0.2$) attacks on MVTec AD, obtained by PLS, APAE and ours.

| Method | Standard AUC | Robust AUC | | | |
|---|---|---|---|---|---|
| | | $l_\infty$-PGD | $l_2$-PGD | $l_\infty$-BPDA | $l_2$-BPDA |
| PLS | 46.4 | 16.0 | 40.8 | - | - |
| APAE | 64.7 | 29.9 | 61.2 | 30 | 61.2 |
| FreeRAD | **95.7** | **80.5** | **88.8** | **88.3** | **89.6** |

### 5.4 DEFENDING AGAINST STRONGER ADAPTIVE ATTACKS

So far we have shown that FreeRAD is indeed robust to PGD and BPDA attacks in Section 5.2 and 5.3. To further verify its robustness in more challenging settings, we test FreeRAD against adaptive attacks where the attacker is assumed to already know about our diffusion model-based anomaly detection method and design attacks against our defense adaptively. Since the diffusion process in our method introduces extra stochasticity, which plays an important role in defending against adversarial perturbations, we consider applying EOT to PGD, which is designed for circumventing randomized defenses. In particular, EOT calculates the expected gradients over the randomization as a proxy for the true gradients of the inference model using Monte Carlo estimation (Athalye et al., 2018b;a; Lee et al., 2022b). We set the number of samples $n = 20$ for the EOT attacks following Nie et al. (2022).

Table 5: Robust AUC against $l_\infty$-PGD, $l_\infty$-EOT-PGD ($\epsilon = 2/255$, EOT=20), and $l_2$-PGD, $l_2$-EOT-PGD attacks ($\epsilon = 0.2$, EOT=20) on *Bottle*, *Grid*, *Toothbrush*, *Wood* from MVTec AD. We also show the difference between the results of PGD and EOT-PGD attacks.

| Categoty | Robust AUC | | diff | Robust AUC | | diff |
|---|---|---|---|---|---|---|
| | $l_\infty$-PGD | $l_\infty$-EOT-PGD | | $l_2$-PGD | $l_2$-EOT-PGD | |
| Bottle | 88.0 | 87.0 | $-1.0$ | 95.5 | 94.3 | $-1.2$ |
| Grid | 99.8 | 99.8 | $-0.0$ | 99.9 | 100 | $+0.1$ |
| Toothbrush | 95.8 | 92.5 | $-3.3$ | 99.4 | 98.1 | $-1.3$ |
| Wood | 95.2 | 86 | $-9.2$ | 95.5 | 94.4 | $-1.1$ |
| **Average** | 95.8 | 91.3 | $-4.5$ | 97.6 | 96.7 | $-0.9$ |

Table 5 shows the robust AUC against EOT-PGD attacks and the difference between the results of standard PGD attacks and EOT-PGD attacks on *Bottle*, *Grid*, *Toothbrush*, *Wood* categories of MVTec AD. We observe that the adversarial robustness is not affected too much by EOT. Specifically, the average robust AUC slightly drops $4.5\%$ and $0.9\%$ compared against standard $l_\infty$-PGD and $l_2$-PGD attacks, respectively. These results suggest that our method has empirically strong robustness against adaptive attacks with EOT. Since other baselines use deterministic inference models, it is unnecessary to apply EOT to evaluate their adversarial robustness.

### 5.5 EXTENSION: CERTIFIED ADVERSARIAL ROBUSTNESS

In this section, we apply randomized smoothing (Cohen et al., 2019) to our diffusion-based anomaly detector and construct a new "smoothed" detector for certified robustness. Given a well-trained FreeRAD detector $A_\theta(\cdot)$ that outputs the anomaly score, we can construct a binary anomaly classi-

fier with any defined threshold $h$:

$$f(x) = \begin{cases} \text{normal,} & \text{if } A_\theta(x) \leq h \\ \text{anomaly,} & \text{otherwise} \end{cases} \tag{5.1}$$

Then we can make predictions by constructing a Gaussian smoothed FreeRAD and compare with $h$. The smoothed FreeRAD enjoys provable robustness, which is summarized in the following theorem:

**Theorem 5.1.** [Smoothed FreeRAD] Given a well-trained FreeRAD detector $A_{\boldsymbol{\theta}}(\mathbf{x})$, for any given threshold $h$ and $\boldsymbol{\delta} \sim \mathcal{N}(0, \sigma^2\mathbf{I})$, if it satisfies $\mathbb{P}[A_{\boldsymbol{\theta}}(\mathbf{x} + \boldsymbol{\delta}) > h] \geq p_{\text{anomaly}}(h) > 1/2$, then $\mathbb{E}_{\boldsymbol{\delta}}[A_{\boldsymbol{\theta}}(\mathbf{x} + \boldsymbol{\delta})] > h$ for all $||\boldsymbol{\delta}||_2 < R(h)$ where $R(h) = \sigma\Phi^{-1}(p_{\text{anomaly}}(h))$. On the other hand, if it satisfies $\mathbb{P}[A_{\boldsymbol{\theta}}(\mathbf{x} + \boldsymbol{\delta}) < h] \geq p_{\text{normal}}(h) > 1/2$, then $\mathbb{E}_{\boldsymbol{\delta}}[A_{\boldsymbol{\theta}}(\mathbf{x} + \boldsymbol{\delta})] < h$ for all $||\boldsymbol{\delta}||_2 < R(h)$ where $R(h) = \sigma\Phi^{-1}(p_{\text{normal}}(h))$.

Theorem 5.1 can be used to certify the robustness of a sample $\mathbf{x}$ given any threshold $h$. The estimation of $p_{\text{normal}}(h)$ and $p_{\text{anomaly}}(h)$ can be done using Monte Carlo sampling similar to Cohen et al. (2019). However, the obtained certified radius is highly related to the threshold $h$. Thus the certified accuracy metric cannot fully represent the quality of the anomaly detection if the inappropriate threshold is selected. To solve this issue, we also propose the new certified AUC metric for measuring the certified robustness performance at multiple distinct thresholds. Specifically, for each threshold candidate, we can make predictions by $\mathbb{E}_{\boldsymbol{\delta}}[A_{\boldsymbol{\theta}}(\mathbf{x}+\boldsymbol{\delta})]$ and compute certified TPR and FPR according to prediction results and their certified radius. After iterating all possible thresholds, we calculate final AUC scores based on the collection of certified TPRs and FPRs on various thresholds.

Table 6 shows the certified robustness achieved by FreeRAD. For example, we achieve $98.2\%$ certified AUC at $l_2$ radius $0.2$ on *gird* sub-dataset, which indicates that there does not exist any adversarial perturbations $\boldsymbol{\delta}$ ($||\boldsymbol{\delta}|| \leq 0.2$) that can make the AUC lower than $98.2\%$. One major limitation of randomized smoothing on anomaly detection tasks is that the noise level can not be much high, otherwise the anomalous features might be covered by the Gaussian noise such that the detector can not distinguish anomalous samples from normal samples. For instance, there is only $12.4\%$ certified AUC on *Bottle* sub-dataset under the noise level $\sigma = 0.25$. The performance gap on different datasets (e.g., $98.2\%$ vs. $12.4\%$) under the same noise level ($\sigma = 0.25$) indicates that the selection of the noise level might depend on specific anomaly features.

Table 6: Certified AUC on *Bottle*, *Grid*, *Toothbrush*, *Wood* datasets from MVTec AD benchmark at varying levels of Gaussion noise $\sigma$.

| Noise | Certified AUC at $l_2$ radius $\epsilon$ | | | | Noise | Certified AUC at $l_2$ radius $\epsilon$ | | | |
|---|---|---|---|---|---|---|---|---|---|
| | 0 | 0.05 | 0.1 | 0.2 | | 0 | 0.05 | 0.1 | 0.2 |
| $\sigma = 0.0625$ | **99.9** | 95.7 | 0 | 0 | $\sigma = 0.0625$ | **100** | 99.9 | 0 | 0 |
| $\sigma = 0.125$ | **99.9** | **97.8** | **92.0** | 0 | $\sigma = 0.125$ | **100** | **100** | **99.9** | 0 |
| $\sigma = 0.25$ | 66.5 | 47.3 | 28.8 | **12.4** | $\sigma = 0.25$ | 99.6 | 99.2 | 98.2 | **98.2** |
| (a) *Bottle* | | | | | (b) *Grid* | | | | |

| Noise | Certified AUC at $l_2$ radius $\epsilon$ | | | | Noise | Certified AUC at $l_2$ radius $\epsilon$ | | | |
|---|---|---|---|---|---|---|---|---|---|
| | 0 | 0.05 | 0.1 | 0.2 | | 0 | 0.05 | 0.1 | 0.2 |
| $\sigma = 0.0625$ | **100** | 98.2 | 0 | 0 | $\sigma = 0.0625$ | **98.5** | 87.9 | 0 | 0 |
| $\sigma = 0.125$ | **100** | **99.4** | 97.2 | 0 | $\sigma = 0.125$ | 98.3 | **94** | **84.8** | 0 |
| $\sigma = 0.25$ | **100** | **99.4** | **98.1** | **91.7** | $\sigma = 0.25$ | 96 | 88.6 | 79.2 | **66.7** |
| (c) *Toothbrush* | | | | | (d) *Wood* | | | | |

## 6 CONCLUSION

Adversarial robustness is a critical factor for the practical deployment of deep anomaly detection models. In this work, we propose an adversarially robust anomaly detector based on the diffusion model that leverages reconstruction error to detect anomalies and utilizes the diffusion process to gradually remove adversarial perturbations for better robustness. We empirically show that our method provides outstanding adversarial robustness while also maintaining strong anomaly detection performance on benchmark datasets. One major advantage of our method is that it gains adversarial robustness for free: the diffusion model functions both as an anomaly detector and an adversarial defender, thus no extra adversarial training or data purification is needed as in standard robust image classification tasks.

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

## A  TRAINING OBJECTIVE OF THE DIFFUSION MODEL

In this section, we introduce the hybrid training objective proposed by (Nichol & Dhariwal, 2021). Specifically, training diffusion models can be performed by optimizing the commonly used variational bound on negative log-likelihood as follows (Ho et al., 2020):

$$L_{\text{vb}} := L_0 + L_1 + \ldots + L_{T-1} + L_T \tag{A.1}$$

$$L_0 := -\log p_{\boldsymbol{\theta}}(\mathbf{x}_0|\mathbf{x}_1) \tag{A.2}$$

$$L_{t-1} := D_{KL}(q(\mathbf{x}_{t-1}|\mathbf{x}_t,\mathbf{x}_0)||p_{\boldsymbol{\theta}}(\mathbf{x}_{t-1}|\mathbf{x}_t)) \tag{A.3}$$

$$L_T := D_{KL}(q(\mathbf{x}_T|\mathbf{x}_0)||p(\mathbf{x}_T)) \tag{A.4}$$

Ho et al. (2020) suggest that directly optimizing this variational bound $L_{vb}$ would produce much more gradient noise during training and propose a reweighted simplified objective $L_{simple}$:

$$L_{\text{simple}} = \mathbb{E}_{t,\mathbf{x}_0,\boldsymbol{\epsilon}}[\|\boldsymbol{\epsilon} - \boldsymbol{\epsilon}_{\boldsymbol{\theta}}(\mathbf{x}_t,t)\|]. \tag{A.5}$$

However, this $L_{simple}$ model suffers from sample quality loss when using a reduced number of denoising steps (Nichol & Dhariwal, 2021). Nichol & Dhariwal (2021) find that training diffusion models via a hybrid objective:

$$L_{\text{hybrid}} = L_{\text{simple}} + \lambda L_{\text{vb}} \tag{A.6}$$

greatly improves its practical applicability by generating high-quality samples with fewer denoising steps, which is helpful for using diffusion models on applications with high-efficiency requirements such as real-time anomaly detection (Sun et al., 2021). In particular, we parameterize the variance term $\boldsymbol{\Sigma}_{\boldsymbol{\theta}}(\mathbf{x}_t,t)$ in Eq.4.1 as an interpolation between $\beta_t$ and $\widetilde{\beta}_t$ in the log domain following (Nichol & Dhariwal, 2021):

$$\boldsymbol{\Sigma}_{\boldsymbol{\theta}}(\mathbf{x}_t,t) = exp(\mathbf{v}\log\beta_t + (1-\mathbf{v})\log\widetilde{\beta}_t) \tag{A.7}$$

where $\mathbf{v}$ is the model output. Following Nichol & Dhariwal (2021), we set $\lambda = 0.001$ and apply a stop-gradient to the $\boldsymbol{\mu}_{\boldsymbol{\theta}}(\mathbf{x}_t,t)$ output for $L_{\text{vb}}$ to prevent $L_{\text{vb}}$ from overwhelming $L_{\text{simple}}$

## B  ADDITIONAL ALGORITHMS

### B.1  ARBITRARY-SHOT ROBUST RECONSTRUCTION IN FREERAD

In this section, we attach the complete algorithm for arbitrary-shot robust reconstruction motivated by (Nichol & Dhariwal, 2021). Given an arbitrary denoising steps $S = \{S_m, S_{m-1}, \ldots, S_1\}(m \leq k, k = S_m > S_{m-1} > \cdots > S_1 >= 1)$, in each step $t \in [1, m]$, we iteratively predict the true point $\mathbf{x}$ given the current diffused data $\mathbf{x}_{S_t}$, termed $\widetilde{\mathbf{x}}_0$, them sampling new iterate $\mathbf{x}_{S_{t-1}}$ according to the current prediction $\widetilde{\mathbf{x}}_0$ and current diffused data $\mathbf{x}_{S_t}$.

---

**Algorithm 3** Arbitrary-shot Robust Reconstruction in FreeRAD

---

**Input:** Test images: $\mathbf{x}$, diffusion steps: $k$, arbitrary generation steps: $S = \{S_m, S_{m-1}, \ldots, S_1\}(m \leq k, k = S_m > S_{m-1} > \cdots > S_1 >= 1)$
**Output:** Reconstructions of $\mathbf{x}$: $\widetilde{\mathbf{x}}$

1: $\mathbf{x}_0 = \mathbf{x}$
2: $\boldsymbol{\epsilon} \sim \mathcal{N}(0, \mathbf{I})$
3: $\mathbf{x}_k = \sqrt{\overline{\alpha}_k}\mathbf{x}_0 + \sqrt{1 - \overline{\alpha}_k}\boldsymbol{\epsilon}$
4: **for** $t = m, \ldots, 1$ **do**                                    ▷ arbitrary-shot denoising
5:     $\widetilde{\mathbf{x}}_0 = \frac{1}{\sqrt{\overline{\alpha}_{S_t}}}(\mathbf{x}_{S_t} - \sqrt{1 - \overline{\alpha}_{S_t}}\boldsymbol{\epsilon}_{\boldsymbol{\theta}}(\mathbf{x}_{S_t}, S_t))$
6:     **if** $t > 1$ **then**
7:         $\mathbf{z} \sim \mathcal{N}(0, \mathbf{I})$
8:         $\mathbf{x}_{S_{t-1}} = \frac{\sqrt{\overline{\alpha}_{S_{t-1}}}\beta_{S_t}}{1-\overline{\alpha}_{S_t}}\widetilde{\mathbf{x}}_0 + \frac{\sqrt{\overline{\alpha}_{S_t}}(1-\overline{\alpha}_{S_{t-1}})}{1-\overline{\alpha}_{S_t}}\mathbf{x}_{S_t} + \sqrt{\boldsymbol{\Sigma}_{\boldsymbol{\theta}}(\mathbf{x}_{S_t}, S_t)}\mathbf{z}$
9:     **end if**
10: **end for**
11: $\widetilde{\mathbf{x}} = \widetilde{\mathbf{x}}_0$

---

## B.2 ANOMALY SCORE CALCULATION

In this section, we attach the complete algorithm for anomaly score calculation. Given test image $\mathbf{x} \in \mathbb{R}^{C \times H \times W}$ and its reconstruction $\widetilde{\mathbf{x}} \in \mathbb{R}^{C \times H \times W}$ obtained by FreeRAD, we first calculate the Multiscale Reconstruction Error Map. In particular, we choose a scale schedule $L = \{1, \frac{1}{2}, \frac{1}{4}, \frac{1}{8}\}$. For each scale $l$, we compute the error map $\mathrm{Err}(\mathbf{x}, \widetilde{\mathbf{x}})_l$ between the downsampled input $\mathbf{x}^l$ and the downsampled reconstruction $\widetilde{\mathbf{x}}^l$ with $\frac{1}{C}\sum_{c=1}^{C}(\mathbf{x}^l - \widetilde{\mathbf{x}}^l)^2_{[c,:,:]}$ where the square operator here refers to element-wise square operation, then unsampled to the original resolution. The final $\mathrm{Err}_{\mathrm{ms}}$ is obtained by averaging each scale's error map and applying a mean filter for better stability similar to Zavrtanik et al. (2021): $\mathrm{Err}_{\mathrm{ms}}(\mathbf{x}, \widetilde{\mathbf{x}}) = (\frac{1}{N_L}\sum_{l \in L}\mathrm{Err}(\mathbf{x}, \widetilde{\mathbf{x}})_l) * f_{s \times s}$ where $f_{s \times s}$ is the mean filter of size $s \times s$, $*$ is the convolution operation. Similar to Pirnay & Chai (2022), we take the pixel-wise maximum of the absolute deviation of the $\mathrm{Err}_{\mathrm{ms}}(\mathbf{x}, \widetilde{\mathbf{x}})$ to the normal training data as the scalar anomaly score.

---

**Algorithm 4** Anomaly Score Calculation in FreeRAD

---

**Input:** Test image: $\mathbf{x} \in \mathbb{R}^{C \times H \times W}$, Reconstructed image: $\widetilde{\mathbf{x}} \in \mathbb{R}^{C \times H \times W}$,
**Output:** Anomaly score: $A(\mathbf{x})$

1: **for** $l$ in $L = \{1, \frac{1}{2}, \frac{1}{4}, \frac{1}{8}\}$ **do**           $\triangleright$ $L$ is a downsampling scale schedule
2:      $\mathbf{x}^l = \mathrm{downsample}(l, \mathbf{x}) \in \mathbb{R}^{C \times (l \times H) \times (l \times W)}$
3:      $\widetilde{\mathbf{x}}^l = \mathrm{downsample}(l, \widetilde{\mathbf{x}}) \in \mathbb{R}^{C \times (l \times H) \times (l \times W)}$
4:      $\mathrm{Err}(\mathbf{x}, \widetilde{\mathbf{x}})_l = \mathrm{upsample}(\frac{1}{l}, \frac{1}{C}\sum_{c=1}^{C}(\mathbf{x}^l - \widetilde{\mathbf{x}}^l)^2_{[c,:,:]}) \in \mathbb{R}^{H \times W}$      $\triangleright$ element-wise square
5: **end for**
6: $\mathrm{Err}_{\mathrm{ms}}(\mathbf{x}, \widetilde{\mathbf{x}}) = (\frac{1}{N_L}\sum_{l \in L}\mathrm{Err}(\mathbf{x}, \widetilde{\mathbf{x}})_l) * f_{s \times s} \in \mathbb{R}^{H \times W}$    $\triangleright$ $f_{s \times s}$ is a mean filter of size $(s \times s)$
7: $A(\mathbf{x}) = \max(|\mathrm{Err}_{\mathrm{ms}}(\mathbf{x}, \widetilde{\mathbf{x}}) - \frac{1}{N_Z}\sum_{z \in Z}\mathrm{Err}_{\mathrm{ms}}(\mathbf{z}, \widetilde{\mathbf{z}})|)$    $\triangleright$ $Z$ is the set of normal training images

---

## C MORE DETAILS OF EXPERIMENTAL SETTINGS

### C.1 HYPERPARAMETERS OF THE DIFFUSION MODEL

The diffusion model in our experiments uses the linear noise schedule (Ho et al., 2020). The number of channels in the first layer is 128, and the number of heads is 1. The attention resolution is $16 \times 16$. We adopt PyTorch as the deep learning framework for implementations. We train the model using Adam optimizer with the learning rate of $10^{-4}$ and the batch size of 2. The model is trained for 30000 iterations for all categories of data. We set diffusion steps $T = 1000$ for training. We list the choice of $k$ for each category as follows:

Table 7: The choices of $k$ for each category of MVTec AD dataset

| k | categories |
|---|---|
| 50 | Capsule, Pill, Bottle, Leather, Tile, Zipper |
| 100 | Grid, Screw, Toothbrush, Wood |
| 200 | Transistor, Hazelnut, Carpet, Cable |
| 300 | Metal Nut |

## D MORE EXPERIMENTAL RESULTS

### D.1 IMPACT OF THE DIFFUSION STEP

Here we first provide anomaly detection performance of proposed FreeRAD on clean data at varying diffusion steps $k$ at inference time. We test with $t \in \{25, 50, 100, 200, 300\}$. As shown in Table 8, different datasets may not have the same optimal $k$. A principle for anomaly detection on clean

data is that k should be chosen such that the amount of Gaussian noise is dominating the anomaly signals while the high-level features of the input data are still preserved for reconstruction. In terms of the adversarial data, $k$ should also be large enough to add sufficient Gaussian noise to dominate adversarial perturbation. As in Table 9, we can see that our method obtains the best performance on clean data at $k = 25$. However, the robust AUC is not satisfying, since the noise added in the diffusion process cannot dominate the adversarial perturbations. Therefore, we can choose larger $k$ (e.g., 50) to obtain better robust performance with slight performance loss on clean data.

Table 8: AUC results on 15 categories from MVTec AD at varying diffusion steps $k$ at inference time

| | Category | $k = 25$ | $k = 50$ | $k = 100$ | $k = 200$ | $k = 300$ |
|---|---|---|---|---|---|---|
| Texture | Carpet | 71.8 | 64.9 | 73.8 | **82.7** | 80.9 |
| | Grid | **100** | **100** | **100** | **100** | **100** |
| | Leather | **100** | **100** | **100** | 99.3 | 98.4 |
| | Tile | **100** | 99.2 | 95.4 | 81.4 | 74.0 |
| | Wood | 95.4 | 98.2 | 98.3 | 97.9 | 97.1 |
| Object | Bottle | 98.9 | **100** | 99.6 | 99.1 | 97.9 |
| | Cable | 78.9 | 78.8 | 79.2 | **79.5** | 77.7 |
| | Capsule | **96.3** | 93.9 | 90.5 | 84.6 | 80.7 |
| | Hazelnut | 95.8 | 96.2 | 97.3 | **97.5** | 96.2 |
| | Metal Nut | 79.5 | 83.8 | 91.0 | 91.3 | **93.5** |
| | Pill | **98** | 97.2 | 94.4 | 86.6 | 68.6 |
| | Screw | 97.3 | 95.0 | **99.3** | 80.8 | 66 |
| | Toothbrush | **100** | **100** | **100** | 99.7 | 99.7 |
| | Transistor | 87.5 | 87.8 | 90.6 | **93.7** | 93.2 |
| | Zipper | **100** | **100** | 99.7 | 96.4 | 95.0 |

Table 9: Standard AUC and robust AUC against $l_2$-PGD attacks ($\epsilon = 0.2$) at varying diffusion step $k$ on *Capsule* and *Pill* from MVTec AD.

| Category | $k = 25$ | | $k = 50$ | |
|---|---|---|---|---|
| | Standard AUC | Robust AUC | Standard AUC | Robust AUC |
| Capsule | 96.3 | 49.5 | 93.9 | 68.1 |
| Pill | 98 | 53.9 | 97.2 | 80.3 |

## D.2 Reducing denoising steps

In this section, we provide the anomaly detection performance of FreeRAD on clean data at varying denoising steps in Table 10 by running Algorithm 3 for reconstruction and using Algorithm 4 to compute anomaly score. Specifically, we test with several denoising steps schedules from one-shot denoising (1-step) to full-shot denoising ($k$-step) and intermediate settings such as $0.05k$, $0.1k$, $0.25k$, and $0.5k$. We can see that one-shot denoising obtains the highest AUC scores on all four datasets. Moreover, we report the inference time (in seconds) at varying denoising steps in Table 11 on an NVIDIA TESLA K80 GPU, where the inference time increases linearly with denoising steps. We show that the inference with one-shot denoising could process a single image in 0.5 seconds, which demonstrates the applicability of our method FreeRAD on real-time tasks. These experimental results clearly indicate that FreeRAD with reconstruction by one-shot denoising achieves both the best detection effectiveness and time efficiency.

## D.3 Comparison with Robust Anomaly Detection methods

In this section, we compare our method FreeRAD with robust anomaly detection methods such as Robust Autoencoder (Zhou & Paffenroth, 2017), which was proposed to handle noise and outlier data points, although the adversarial perturbation was not explicitly considered in their work. Table 12 clearly shows that our method still largely outperforms RAE no both clean data and adversarial data.

Table 10: AUC results on *Screw*, *Toothbrush*, *Wood*, *Transistor* at varying denoising steps. The choice of $k$ for each category follows Table 7.

| Category | $1-$step | $0.05k$-step | $0.1k$-step | $0.25k$-step | $0.5k$-step | $k$-step |
|---|---|---|---|---|---|---|
| Screw | **99.3** | 97.3 | 96.8 | 97.2 | 95.1 | 96.4 |
| Toothbrush | **100** | 99.7 | **100** | 99.4 | **100** | **100** |
| Transistor | **93.7** | 92.3 | 87.2 | 89.5 | 82.1 | 86.2 |
| Wood | **98.3** | 97.1 | 98.2 | 95.9 | 98.2 | 96 |

Table 11: Inference time (in seconds) for a single image on *Toothbrush* and *Transistor* by varying denoising steps, where the inference time increases over one-shot denoising is given in parenthesis.The choice of $k$ for each category follows Table 7.

| Category | 1-step | $0.05k$-step | $0.1k$-step | $0.25k$-step | $0.5k$-step | $k$-step |
|---|---|---|---|---|---|---|
| Toothbrush | 0.5 | $2.28_{(\times 4.6)}$ | $4.63_{(\times 9.3)}$ | $11.53_{(\times 23.6)}$ | $23.03_{(\times 46.1)}$ | $46.06_{(\times 92.1)}$ |
| Transistor | 0.5 | $4.98_{(\times 10)}$ | $10.01_{(\times 20)}$ | $25_{(\times 50)}$ | $50_{(\times 100)}$ | $99.78_{(\times 199.6)}$ |

Table 12: Average standard AUC and robust AUC against $l_\infty$-PGD($\epsilon = 2/255$), $l_2$-PGD($\epsilon = 0.2$)) attacks on MVTec AD, obtained by RAE and ours.

| Method | Standard AUC | Robust AUC | |
|---|---|---|---|
| | | $l_\infty$-PGD | $l_2$-PGD |
| RAE | 57.1 | 16.8 | 49.8 |
| FreeRAD | **95.7** | **80.5** | **88.8** |

## D.4 DEFENDING AGAINST AUTOATTACK

We have shown that FreeRAD is robust to adaptive attacks EOT-PGD in Section 5.4. In this section, we incorporate additional strong attack baselines, AutoAttack (Croce & Hein, 2020) which ensemble multiple white-box and black-box attacks such as APGD attacks and Square attacks. Specifically, we used two versions of AutoAttack: (i) standard AutoAttack and (ii) random AutoAttack (EOT+AutoAttack), which is used for evaluating stochastic defense methods. We summarize the standard AUC and robust AUC of our proposed FreeRAD in the following Table 13. The robust AUC scores of FreeRAD against AutoAttack are still largely higher than other SOTAs against relatively weaker PGD attacks as shown in Table 2 and 3, thus there is no need to evaluate other methods' robustness against stronger AutoAttack.

Table 13: Standard AUC and robust AUC against $l_\infty$-AutoAttack($\epsilon = 2/255$), $l_2$-AutoAttack($\epsilon = 0.2$)) on *Bottle*, *Grid*, *Toothbrush*, *Wood* from MVTec AD

| Category | Standard AUC | Robust AUC | | | |
|---|---|---|---|---|---|
| | | $l_\infty$-standard AA | $l_2$-standard AA | $l_\infty$-random AA | $l_2$-random AA |
| Bottle | 100 | 76.5 | 87.8 | 73.4 | 87.6 |
| Grid | 100 | 98.2 | 99.2 | 98.2 | 98.8 |
| Toothbrush | 100 | 73.6 | 84.2 | 65.8 | 86.1 |
| wood | 99.8 | 72.3 | 75.2 | 64.8 | 75.3 |
| **Average** | 100 | 80.1 | 86.6 | 75.6 | 87.0 |

### D.5 EXPERIMENTS ON NOVELTY DETECTION DATASET

Novelty Detection (i.e., semantic anomaly detection) refers to the problem of determining if test data is from the known class (normal) or novel class (anomalous) (Yang et al., 2021). We perform experiments with novelty detection on the CIFAR-10 dataset (Krizhevsky et al., 2009) which has 10 categories with 60000 natural images. Under the setting of novelty detection, one category is regarded as a known class, and other categories are considered novel classes. Hence we train the corresponding model for each category respectively. We evaluate and compare our proposed FreeRAD with several SOTA methods that include FastFlow (Yu et al., 2021), and CFA (Lee et al., 2022a). We summarize the standard AUC and robust AUC against $l_\infty$-PGD and $l_2$-PGD attacks in Table 14. The results show that our method still largely outperforms the baselines method regarding robust AUC while maintaining a strong novelty detection performance on clean data.

Table 14: Standard AUC (in parenthesis) and robust AUC against $l_\infty$-PGD attacks ($\epsilon = 2/255$) and $l_2$-PGD attacks ($\epsilon = 0.2$) on CIFAR-10 dataset, obtained by different state-of-the-art anomaly detectors and ours.

| Attacks | $l_\infty$-PGD | | | $l_2$-PGD | | |
|---|---|---|---|---|---|---|
| Category | FastFlow | CFA | FreeRAD(Ours) | FastFlow | CFA | FreeRAD(Ours) |
| Bird | $0.3^{(63.0)}$ | $0.3^{(68.1)}$ | $\mathbf{57.9}^{(71.9)}$ | $2.1^{(63.0)}$ | $1.7^{(68.1)}$ | $\mathbf{62.2}^{(71.9)}$ |
| Plane | $4.1^{(74.2)}$ | $1.2^{(71.8)}$ | $\mathbf{54.9}^{(79.3)}$ | $10.3^{(74.2)}$ | $3.5^{(71.8)}$ | $\mathbf{64.8}^{(79.3)}$ |
| Car | $1.9^{(81.7)}$ | $0.0^{(76.3)}$ | $\mathbf{50.6}^{(70.4)}$ | $6.9^{(81.7)}$ | $3.9^{(76.3)}$ | $\mathbf{61.7}^{(70.4)}$ |
| Cat | $0.2^{(45.6)}$ | $0.3^{(58.7)}$ | $\mathbf{28.7}^{(56.7)}$ | $0.8^{(45.6)}$ | $1.3^{(58.7)}$ | $\mathbf{38.9}^{(56.7)}$ |
| Deer | $0.5^{(56.1)}$ | $1.0^{(74.6)}$ | $\mathbf{60.5}^{(71.5)}$ | $1.6^{(56.1)}$ | $5.4^{(74.6)}$ | $\mathbf{63.2}^{(71.5)}$ |
| Dog | $1.0^{(72.7)}$ | $1.0^{(64.5)}$ | $\mathbf{39.1}^{(56.4)}$ | $3.7^{(72.7)}$ | $3.0^{(64.5)}$ | $\mathbf{45.4}^{(56.4)}$ |
| Frog | $0^{(79.7)}$ | $0.9^{(81.7)}$ | $\mathbf{59.6}^{(71.3)}$ | $1.5^{(79.7)}$ | $5.2^{(81.7)}$ | $\mathbf{62.5}^{(71.3)}$ |
| Horse | $1.2^{(76.4)}$ | $1.6^{(74.9)}$ | $\mathbf{47.1}^{(60.4)}$ | $4.6^{(76.4)}$ | $5.0^{(74.9)}$ | $\mathbf{51.7}^{(60.4)}$ |
| Ship | $1.8^{(81.2)}$ | $1.3^{(81.0)}$ | $\mathbf{65.2}^{(77.5)}$ | $8.0^{(81.2)}$ | $5.6^{(81.0)}$ | $\mathbf{68.9}^{(77.5)}$ |
| Truck | $3.7^{(83.7)}$ | $0.4^{(74.8)}$ | $\mathbf{23.2}^{(45.0)}$ | $13.1^{(83.7)}$ | $3.9^{(74.8)}$ | $\mathbf{28.5}^{(45.0)}$ |
| **Average** | $1.5^{(71.4)}$ | $0.8^{(72.6)}$ | $\mathbf{48.7}^{(66.0)}$ | $5.3^{(71.4)}$ | $3.9^{(72.6)}$ | $\mathbf{54.8}^{(66.0)}$ |

## E    MORE ANALYSIS OF THE ADVERSARIAL ROBUSTNESS OF DDPMS

In this section, we provide more theoretical analysis on the defense mechanism of DDPMs. Since the adversarial perturbations would be dominated by the added noise from the diffusion process, such that the clean data distribution and adversarially perturbed data distribution get closer. Intuitively, after performing the full diffusion process, any data would converge to pure standard Gaussian as mentioned in Section 4.1. This suggests that the tiny adversarial perturbations will be gradually washed out and have little effect on the final output after denoising. Moreover, the stochasticity introduced from the sampling in the diffusion process (Eq.4.2) makes it well-suited for combining with randomized smoothing strategies and building certified robustness without much loss on anomaly detection performances. The following theorem confirms that the diffusion process in DDPMs could make the KL-divergence of diffused clean data distribution and diffused adversarially perturbed data distribution decreases gradually.

**Theorem E.1.** Given any clean data distribution $p(\mathbf{x})$ and adversarially perturbed data distribution $q(\mathbf{x})$, we denote by $p_t$ the distribution of $\mathbf{x}_t$ derived from the t-step diffusion process in Eq. 4.2 when $\mathbf{x}_0 \sim p(\mathbf{x})$. Accordingly, we denote by $q_t$ the distribution of $\mathbf{x}_t$ derived from the t-step diffusion process when $\mathbf{x}_0 \sim q(\mathbf{x})$. If $t \in [0, T]$ and $T \to \infty$, the diffusion process in DDPM converges to a continuous process and

$$\frac{\partial D_{KL}(p_t||q_t)}{\partial t} \le 0,$$

i.e., the KL-divergence of $p_t$ and $q_t$ monotonically decreases during the diffusion process.

*Proof:* The proof mainly follow from Song et al. (2021); Nie et al. (2022).

Following derivations from Song et al. (2021), the discrete Markov chain used in DDPM $\mathbf{x}_i = \sqrt{1-\beta_i}\mathbf{x}_{i-1} + \sqrt{\beta_i}\boldsymbol{\epsilon}_{i-1}, i = 1, \cdots, T$, can be re-written as

$$\mathbf{x}_i = \sqrt{1 - \frac{\overline{\beta}_i}{T}}\mathbf{x}_{i-1} + \sqrt{\frac{\overline{\beta}_i}{T}}\boldsymbol{\epsilon}_{i-1}, \quad i = 1, \cdots, T \tag{E.1}$$

where $\overline{\beta}_i = T\beta_i$. When $T \to \infty$, $\overline{\beta}_i$ becomes a function $\beta(t)$ indexed by $t \in [0, 1]$. Denote $\overline{\beta}_i$, $\mathbf{x}_i$, and $\boldsymbol{\epsilon}_i$ as $\beta(\frac{i}{T})$, $\mathbf{x}(\frac{i}{T})$, and $\boldsymbol{\epsilon}(\frac{i}{T})$, respectively, we can rewrite Eq. E.1 as below:

$$\begin{aligned}
\mathbf{x}(t + \Delta t) &= \sqrt{1 - \beta(t + \Delta t)\Delta t}\mathbf{x}(t) + \sqrt{\beta(t + \Delta t)\Delta t}\boldsymbol{\epsilon}(t) \\
&\approx \mathbf{x}(t) - \frac{1}{2}\beta(t + \Delta t)\Delta t\mathbf{x}(t) + \sqrt{\beta(t + \Delta t)\Delta t}\boldsymbol{\epsilon}(t) \\
&\approx \mathbf{x}(t) - \frac{1}{2}\beta(t)\Delta t\mathbf{x}(t) + \sqrt{\beta(t)\Delta t}\boldsymbol{\epsilon}(t),
\end{aligned} \tag{E.2}$$

where $\Delta t = \frac{1}{T}, t \in \{0, \frac{1}{N}, \cdots, \frac{T-1}{T}\}$, and the approximate equality holds when $\Delta t \ll 1$. Hence in the limit of $T \to \infty$ and $\frac{1}{T} \to 0$, Eq. E.2 converges to a continuous time SDE:

$$d\mathbf{x} = -\frac{1}{2}\beta(t)\mathbf{x}dt + \sqrt{\beta(t)}d\mathbf{w}.$$

Let us denote $f(\mathbf{x}, t) := -\frac{1}{2}\beta(t)\mathbf{x}$ and $g(t) := \sqrt{\beta(t)}$. Following the same proof as in Theorem 3.1 in Nie et al. (2022), we have

$$\frac{\partial D_{KL}(p_t||q_t)}{\partial t} = -\frac{1}{2}g^2(t)D_F(p_t||q_t),$$

where $D_F(p_t||q_t) := \int p_t(\mathbf{x})||\log p_t(\mathbf{x}) - \log q_t(\mathbf{x})||^2 d\mathbf{x} \geq 0$ and $D_F(p_t||q_t) = 0$ iff $p_t = q_t$, thus we have

$$\frac{\partial D_{KL}(p_t||q_t)}{\partial t} \leq 0.$$

