# OpenReview forum: "Robustness for Free: Adversarially Robust Anomaly Detection Through Diffusion Model"
_ICLR.cc/2023/Conference — Submitted to ICLR 2023_

### Official Review · Reviewer_sKar · 2022-10-17

**Confidence:** 3
**Clarity, Quality, Novelty And Reproducibility:** This paper is well-written and shows …
**Correctness:** 3
**Technical Novelty And Significance:** 3
**Empirical Novelty And Significance:** 3
**Recommendation:** 5

**Strength And Weaknesses:**

Strength

- The paper has an interesting finding.
- The proposed implementation trick can achieve real-time anomaly detection.

Weakness
- On the clean data, FreeRAD cannot beat the baseline CFA.
-  I think more attacking approaches can be incorporated for evaluation.
- Currently, a few diffusion model-based anomaly detection approaches have been developed. I am curious whether the existing diffusion model-based anomaly detection models are also robust to adversarial attacks.  By further examining the following approaches, we may know whether the adversarial robustness is from the diffusion model or the implementation ideas developed in this paper.

   - Wyatt, Julian, et al. "AnoDDPM: Anomaly Detection With Denoising Diffusion Probabilistic Models Using Simplex Noise." Proceedings of the IEEE/CVF Conference on Computer Vision and Pattern Recognition Workshops. 2022.
   - Wolleb, J., Bieder, F., Sandkühler, R., Cattin, P.C. (2022). Diffusion Models for Medical Anomaly Detection. In Medical Image Computing and Computer Assisted Intervention – MICCAI 2022.

**Summary Of The Paper:**

This paper develops an adversarially robust anomaly detection through the diffusion model, called FreeRAD. This paper makes an interesting finding that leveraging the diffusion model for anomaly detection can achieve adversarial robustness for free. Experimental results show that the model can still achieve good performance on adversarial samples assembled by PGD and BPDA.

**Summary Of The Review:**

I like the interesting finding in this paper but believe more experimental results could make this paper stronger.

---

> ### Author Response · Authors · 2022-11-14
> **Response to reviewer sKar**
>
> Thank you for your suggestions and comments! Please see the clarification and additional experiment results as follows.
>
> ***
>
> **Q1:** On the clean data, FreeRAD cannot beat the baseline CFA.
>
> **A1:** The standard AUC score of FreeRAD on clean data is only slightly lower than the baseline CFA (95.7% VS. 99.2%), while the robust AUC score against adversarial attacks largely outperforms CFA (80.5% VS. 0.2% against $l_{\infty}$ attacks and 88.8% VS. 43.9% against $l_{2}$ attacks). Additionally, several SOTAs such as CFA and FastFlow use extra training data which our method doesn't need.
>
> **Q2:** I think more attacking approaches can be incorporated for evaluation.
>
> **A2:** Thanks for your suggestion. We have incorporated additional strong attack baselines, AutoAttack [1] which ensembles multiple white-box and black-box attacks such as APGD attacks and Square attacks. Specifically, we used two versions of AutoAttack: (i) standard AutoAttack and (ii) random AutoAttack (EOT+AutoAttack), which is used for evaluating stochastic defense methods.  We summarize the standard AUC and robust AUC of our proposed FreeRAD in the following table.  The robust AUC scores of FreeRAD against AutoAttack are still largely higher than other SOTAs against relatively weaker PGD attacks as shown in Table 2 and 3 in the paper, thus there is no need to evaluate other methods' robustness against stronger AutoAttack.
>
> | Category | Standard AUC | Robust AUC ($l_{\infty}$-standard AA) | Robust AUC ($l_{2}$-standard AA)  | Robust AUC ($l_{\infty}$-random AA) | Robust AUC ($l_{2}$-random AA)  |
> | ---- | ---- | ---- | ---- | ---- | ---- |
> | Bottle | 100 | 76.5 | 87.8 | 73.4 | 87.6 |
> | Grid | 100 | 98.2| 99.2 | 98.2 | 98.8 |
> | Toothbrush | 100 | 73.6 | 84.2 | 65.8 | 86.1 |
> | Wood | 99.8| 72.3| 75.2 | 64.8 | 75.3 |
> | Average | 100| 80.1 | 86.6 | 75.6 | 87.0 |
>
>
> [1] Croce, Francesco, and Matthias Hein. "Reliable evaluation of adversarial robustness with an ensemble of diverse parameter-free attacks." _International conference on machine learning_. PMLR, 2020.
>
> **Q3:** Further examining the existing diffusion model-based anomaly detection models
>
> **A3:** Thanks for providing the references. Please note that we also discussed them in our paper (section 2). Here we clarify that even though the two papers are also about anomaly detection by diffusion models, their scenarios are different from ours: (1) They focus on pixel-level anomaly detection (i.e., anomaly localization), and they do not compute image-level anomaly score. While our task here concentrates on image-level anomaly detection, which makes it hard to directly compare the performance; (2) The paper _Diffusion Models for Medical Anomaly Detection_ is under supervised settings which is distinct from our unsupervised settings.  Hence it is unsuitable to directly evaluate the robustness of their models in our scenarios. We have added further discussions in the revision to reflect this.

---

> ### Author Response · Authors · 2022-11-17
> **A friendly reminder of the rebuttal conclusion**
>
> We would like to thank you for your valuable comments. We have responded to your question and provided additional experiment results, and hope it could help address your concerns. In addition, we are more than happy to discuss and address any further questions.

---

### Official Review · Reviewer_4wM3 · 2022-10-23

**Confidence:** 4
**Clarity, Quality, Novelty And Reproducibility:** The paper is clearly written and easy…
**Correctness:** 3
**Technical Novelty And Significance:** 3
**Empirical Novelty And Significance:** 3
**Recommendation:** 6

**Strength And Weaknesses:**

1. The use of diffusion models directly as an anomaly detector (and achieves adversarial robustness) seems novel
2. The paper is clearly written, and easy to follow
3. There seems to have systematic experiments to validate the claim

On the other hand, my main objection at this point is that it seems to be a direct application of diffusion models without any modification (so it is like -- ``surprise, diffusion models also have these nice properties!''). For example, Algorithm 1 seems to be just the reverse process in diffusion models (inlining the sampling algorithm?)



**Summary Of The Paper:**

This paper tries to construct an adversarially robust anomaly detector (note that this is different from constructing an classifier because we only need to do a binary classification of anomaly or not) directly using the diffusion models. The idea is to directly use the reconstruction score in diffusion model to indicate whether this is an anomaly or not -- and since reconstruction is resilient to adversarial perturbations, we don't need extra work

**Summary Of The Review:**

This paper seems a solid work, but at this point it reads to me more like ``nice properties of diffusion models -- it can be directly applied as an anomaly detector that also achieves robustness''. If so, please clarify this in the paper. If not -- please clearly discuss what are the revisions to the existing diffusion models.

---

> ### Author Response · Authors · 2022-11-14
> **Response to reviewer 4wM3**
>
> Thank you for your comments on our work! Please find some clarification as follows.
>
> ***
>
> **Q1:**  nice properties of diffusion models/revisions to the existing diffusion models
>
> **A1:** Please note that our main contribution in this paper is achieving robust anomaly detection which is non-trivial and not accomplished by prior works. Indeed we do not modify the diffusion models themselves, so our proposed method does rely on nice properties of diffusion models. However, we are the first to systematically study how diffusion models can be used to achieve adversarially robust anomaly detection while previously we only know it is a reconstruction-based method that can be used for anomaly detection.
>
> Additionally, we extend our method for certified robustness through randomized smoothing which provides robustness guarantees (Section 5.5). Moreover, the experiment results in Appendix D.2 demonstrate the applicability of our method FreeRAD with reduced denoising (reverse sampling) steps on real-time tasks, which is critical for the practical deployment of adversarially robust anomaly detection. These certainly do not come for free with the diffusion models and are the key component for the success of FreeRAD.

---

> ### Author Response · Authors · 2022-11-17
> **A friendly reminder of the rebuttal conclusion**
>
> We sincerely thanks for your precious comments. Since we are closing to the end date of the open discussion, we would like to kindly remind you to read our response which hopefully should resolve all your concerns. We would love to hear your feedback and also love to answer if you have additional questions.

---

> ### Author Response · Authors · 2022-12-05
> **We sincerely hope the reviewer finds our response helpful.**
>
> Thank you again for your insightful comments. We hope our responses could be helpful in addressing your concerns. If you still have any questions, we are more than happy to discuss them further. Thank you.

---

### Official Review · Reviewer_BFFe · 2022-10-25

**Confidence:** 4
**Correctness:** 4
**Technical Novelty And Significance:** 2
**Empirical Novelty And Significance:** 3
**Recommendation:** 6

**Clarity, Quality, Novelty And Reproducibility:**

The illustration of methodology is clear, but the explanation of experimental results is not so clear. The reason why certain defense-enabled anomaly detectors are chosen and why compare the performance against adaptive attack in such a way need a better explanation. The overall quality is relatively good. The methodology is reasonable and the experiments are well-designed to show the advantages. However, the novelty is not obvious, because there already exist methods leveraging the diffusion model to remove the perturbations which is also the key to calculating the anomaly reconstruction score in this paper. The authors provide detailed algorithms and experimental settings which may help reproduce.


**Strength And Weaknesses:**

Strength:
This paper is well-written and easy to follow. The proposed method takes advantage of the smoothing property of the diffusion model and reaches robust anomaly detection. Experiments on widely-used anomaly detection benchmarks are conducted, and the performance is compared with SOTAs to better illustrate the performance of the proposed method.

Weakness:
- Some experimental results are not very convincing, especially in table 1, where most of the SOTAs fail under PGD attack, but not sure if this is due to the lack of parameter tuning.
- The idea of applying the diffusion model is not very novel and is similar to the idea in the paper Diffusion Models for Adversarial Purification which also uses the diffusion model to remove adversarial perturbations.
- The comparison with defense-enabled anomaly detectors is not very convincing, and the comparison with more methods should be added.
- This paper does not involve other robust anomaly detection methods in the experiment.
- The experiment of defending against the adaptive attack needs more clarification.


**Summary Of The Paper:**

This paper focuses on anomaly detection models and proposes an adversarially robust anomaly detector based on the diffusion model. In detail, they first add gaussian to the image through the forward process, and then reconstruct it through the reverse process;  the Multiscale Reconstruction Error Map is computed as the anomaly score. In the experimental part, they compare the proposed method with SOTAs to show the advantage and combine with randomized smoothing to reach some certified robustness.


**Summary Of The Review:**

This paper leverages the diffusion model to create a robust anomaly detector and provides experimental results to support their claim. It is well-written and the illustration of the methodology is clear. The novelty is not obvious, and the experimental results need a better explanation otherwise it will confuse the reader.

---

> ### Author Response · Authors · 2022-11-14
> **Response to reviewer BFFe**
>
> Thank you for your suggestions and comments! Please find some clarification and additional experiment results as follows.
>
> ***
>
> **Q1:** Most of the SOTAs fail under PGD attack, but not sure if this is due to the lack of parameter tuning.
>
> **A1:** Please note that we have tried our best to ensure all baselines have properly chosen parameter settings. For CFlow method, we directly use the pre-trained weights released by the authors and thus it already enjoyed optimal parameter settings. For other baselines, we have carefully tuned their parameters. The fact that their AUC scores on clean data reproduced by us are very high and close to the results reported in their own papers simply shows that those parameters are well-tuned.
>
>
>
> **Q2:** The idea is similar to the paper Diffusion Models for Adversarial Purification which also uses the diffusion model to remove adversarial perturbations.
>
> **A2:** As highlighted in our paper, we want to emphasize again that our design is fundamentally different from the purification-based adversarial robust models, i.e., _Diffusion Models for Adversarial Purification_ paper, where an extra external purifier (e.g., diffusion model) is needed before the actual classifier for robust classification. Here we don’t have an extra external purifier before the anomaly detector. The diffusion model in our design acts both as an anomaly detector and an adversarial defender.  In fact, directly apply the strategy in “Diffusion Models for Adversarial Purification”, i.e., using the diffusion model as a purifier before another anomaly detector will not work, as the diffusion-based purifier will break the anomaly signals and make the downstream anomaly detector fail to work.
>
>
>
> **Q3:** The comparison with more defense-enabled methods should be added.
>
> **A3:** Thanks for your suggestion!  To the best of our knowledge, PLS [1] and APAE [2]  are the only two defense-enabled anomaly detectors. The comparison with APAE [2] has been reported in the original paper. Here we further added the comparison with PLS [1]. We summarize the standard AUC and robust AUC in the following table. Since the inference process of PLS [1] is differentiable and thus does not cause obfuscated gradients, we don't need to consider BPDA attacks on it. We can clearly observe that FreeRAD outperforms other defense-enabled methods under all attacks. Furthermore, our method also enjoys a significant advantage in terms of standard AUC on clean data.
>
> | Methods| Standard AUC | Robust AUC ( $l_{\infty}$-PGD) | Robust AUC ($l_{2}$-PGD) | Robust AUC ($l_{\infty}$-BPDA) | Robust AUC ($l_{2}$-BPDA) |
> | ---- | ---- | ---- | ---- | ---- | ---- |
> | PLS | 46.4 | 16.0 | 40.8 | - | - |
> | APAE | 64.7 | 29.9 | 61.2 | 30 | 61.2 |
> | FreeRAD(ours) | 95.7 | 80.5 | 88.8 | 88.3 | 89.6 |
>
>
>
> [1] Shao-Yuan Lo, Poojan Oza, and Vishal M Patel. Adversarially robust one-class novelty detection. IEEE Transactions on Pattern Analysis and Machine Intelligence, 2022.
>
> [2] Adam Goodge, Bryan Hooi, See-Kiong Ng, and Wee Siong Ng. Robustness of autoencoders for
> anomaly detection under adversarial impact. In IJCAI, pp. 1244–1250, 2020.
>
>
>
> **Q4:** This paper does not involve other robust anomaly detection methods in the experiment.
>
> **A4:** Thanks for the suggestion! We conjecture that by “robust anomaly detection”, you are referring to methods such as Robust Autoencoder, which was proposed to handle noise/outlier data points (please let us know if you actually refer to other robust anomaly detection methods). Although the adversarial perturbation was not explicitly considered in these works, we agree that it might be interesting to also compare with such robust anomaly detection method RAE [3]. We have conducted the experiments and the following table shows that our method still largely outperforms RAE [3] on both clean data and adversarial data.
>
> | Methods | Standard AUC | Robust AUC ($l_{\infty}$-PGD) | Robust AUC ($l_{2}$-PGD) |
> | ---- | ---- | ---- | ---- |
> | RAE | 57.1 | 16.8 | 49.8 |
> | FreeRAD(ours) | 95.7 | 80.5| 88.8 |
>
>
> [3] Zhou, Chong, and Randy C. Paffenroth. "Anomaly detection with robust deep autoencoders." *Proceedings of the 23rd ACM SIGKDD international conference on knowledge discovery and data mining*. 2017.
>
>
> **Q5:** The experiment of defending against the adaptive attack needs more clarification.
>
> **A5:** We are sorry for the confusion! The adaptive attack means that suppose the attackers know that we are using diffusion-model based anomaly detection, can they design an adaptive attack strategy specifically for our method. Intuitively speaking, the diffusion process (Eq. 4.2) in our method introduces extra stochasticity which plays an important role in defending against adversarial perturbations. Thus we conjecture that attackers might try to eliminate such stochasticity to counter our defense, which naturally leads to the EOT-PGD attack which averages the results over multiple samples for circumventing randomized defenses.

---

> > ### Comment · Reviewer_BFFe · 2022-11-23
> > **Response to the rebuttal**
> >
> > Thank the authors for providing more experiment results and clarification. I still have one concern. Since your method does not rely on an external pre-trained diffusion model, what model do you apply when predicting noise during the reconstruction? Do you train one on the whole training set which contains anomalous signals? If so, will your training process overfit the anomalous signals which lead to the small gap between reconstruction and original one?

---

> > > ### Author Response · Authors · 2022-11-24
> > > **Response to your new question**
> > >
> > > Thanks for your question. Indeed, we trained one DDPM for each training dataset, yet the DDPM is only trained on normal samples (not because we choose to train on normal samples, but because the whole training dataset only has normal samples). Please note that many visual anomaly detection datasets including the MVTec dataset [1] used in our work, are collected in this way (training data is all normal while the test data has normal samples and anomalies). Also as mentioned in [1], collecting anomaly-free data is preferable in practice as only a limited amount of anomalies are available in contrast to normal samples. Existing works on visual anomaly detection tasks, such as [2] [3] also followed such anomaly-free training data setting. We also followed them and trained one DDPM on each training dataset which only contains anomaly-free data, thus the training process wouldn’t overfit the anomalous signals.
> > >
> > > [1] Paul Bergmann, Kilian Batzner, Michael Fauser, David Sattlegger, and Carsten Steger. The mvtec anomaly detection dataset: a comprehensive real-world dataset for unsupervised anomaly detection. International Journal of Computer Vision, 129(4):1038–1059, 2021.
> > >
> > > [2] Sungwook Lee, Seunghyun Lee, and Byung Cheol Song. Cfa: Coupled-hypersphere-based feature adaptation for target-oriented anomaly localization. arXiv preprint arXiv:2206.04325, 2022a.
> > >
> > > [3] Denis Gudovskiy, Shun Ishizaka, and Kazuki Kozuka. Cflow-ad: Real-time unsupervised anomaly detection with localization via conditional normalizing flows. In Proceedings of the IEEE/CVF Winter Conference on Applications of Computer Vision, pp. 98–107, 2022.

---

> > > > ### Comment · Reviewer_BFFe · 2022-12-06
> > > > **Response to authors**
> > > >
> > > > Thanks for your clarification. I will raise my score.

---

> > > > > ### Author Response · Authors · 2022-12-06
> > > > > **Thank you**
> > > > >
> > > > > We are glad that our responses have addressed your concerns. Thank you for raising the score.

---

> > > ### Author Response · Authors · 2022-12-05
> > > **We sincerely hope the reviewer finds our response helpful.**
> > >
> > > Thanks again for your thoughtful comments. We sincerely hope the reviewer find our response useful and update the score if your concerns have been resolved. We are also open to further discussion if there are further questions.

---

> ### Author Response · Authors · 2022-11-17
> **A friendly reminder of the rebuttal conclusion**
>
> We sincerely appreciate your detailed feedback on our work. We have responded to each of your concerns and provided additional experiment results accordingly. Please let us know if your concerns have been addressed. If you have any further questions, we are more than happy to address them before the conclusion of the rebuttal phase.

---

### Official Review · Reviewer_u5qy · 2022-10-31

**Confidence:** 4
**Correctness:** 3
**Technical Novelty And Significance:** 3
**Empirical Novelty And Significance:** 3
**Recommendation:** 5

**Clarity, Quality, Novelty And Reproducibility:**

+ The paper is written well and easy-to-follow.
+ Overall, the paper is placed well into the existing literature.
+ There are some other first works on using diffusion models for AD, but this class of models has not yet been explored much.
+ The paper seems to present all relevant details for reproducibility in the main paper + appendix

---

*Additional Comments*
* Why do Table 5 and Table 6 only consider 4 classes (Bottle, Grid, Toothbrush, Wood)?
* p.1: "Recently, deep learning (DL) based anomaly detection methods have achieved remarkable improvement over traditional anomaly detection strategies." You might reference recent reviews here for further reading, e.g. Ruff et al. (2021) or Pang et al. (2021).

Ruff, Lukas, et al. "A unifying review of deep and shallow anomaly detection." Proceedings of the IEEE 109.5 (2021): 756-795.

Pang, Guansong, et al. "Deep learning for anomaly detection: A review." ACM Computing Surveys (CSUR) 54.2 (2021): 1-38.

**Strength And Weaknesses:**

*Strengths*
+ The paper presents a coherent framework for adversarial attacks on anomaly detectors (targeting both anomalies to appear normal and normal data points to appear anomalous) and evaluates several existing methods w.r.t. their adversarial robustness, finding that many existing methods suffer from not being adversarially robust.
+ The experiments show that the proposed method achieves strong adversarial robustness while maintaining a comparable state-of-the-art detection performance on MVTec-AD.

*Weaknesses*
- I find the main weakness of the paper to lie in the experimental evaluation, which only considers *one* dataset (MVTec-AD). This strongly limits evidence for some of the (fairly general) made claims in the paper. Defect detection on MVTec-AD, where anomalies are rather subtle and low-level, is quite distinct from other visual anomaly detection tasks (e.g. detecting novel classes).
This is (implicitly) acknowledged in the paper: "Note that $k$ should be chosen such that the amount of Gaussian noise is dominating the adversarial perturbations and anomaly signals while the high-level features of the input data are still preserved for reconstruction."
Yet, there is no broader discussion on this point, which I think would be critical.

**Summary Of The Paper:**

This paper proposes an anomaly detection (AD) method based on the diffusion model. Training a diffusion model on unlabeled data (mostly assumed to be normal), the reconstruction error is proposed as an anomaly score (similar to using autoencoders for AD). In addition, using the diffusion model to "diffuse" an input (i.e. adding noise by partially applying the concatenated diffusion model maps) and afterwards reconstructing it, the model is also used for mitigating adversarial perturbations. An experimental evaluation on MVTec-AD is presented showing that the proposed method achieves a similar anomaly detection performance as existing state-of-the-art methods while demonstrating an improved adversarial robustness.

**Summary Of The Review:**

I enjoyed reading this paper and think the presented contributions w.r.t. adversarially robust anomaly detection are interesting and relevant, but the paper is strongly limited in only evaluating on a single dataset which is rather specific. Claims such as in the Conclusion ("We empirically show that our method provides outstanding adversarial robustness while also maintaining strong anomaly detection performance on benchmark datasets.") are simply too far-fetched in my opinion for the provided evidence, which should be corrected for (either by adding more datasets or limiting claims to MVTec-AD).

---

> ### Author Response · Authors · 2022-11-14
> **Response to reviewer u5qy**
>
> Thank you for your suggestions and comments! Please see the response and clarification as follows.
>
> ***
>
> **Q1:** Experiments on only one dataset/new experiments with novel classes detection
>
> **A1:** Thank you for your suggestion. Though our method is not directly designed for novel class detection task but we believe FreeRAD can also function as an adversarially robust novelty detector. Specifically, we evaluate and compare our proposed FreeRAD with other SOTA methods on CIFAR10 for detecting novel classes. We summarize the standard AUC (in parenthesis) and robust AUC against $l_{\infty}$-PGD and $l_2$-PGD attacks in the following tables, respectively (due to time constraint, we only present two baselines here: FastFlow and CFA. We will continue working on it and provide results on all baselines in the camera-ready). These results show that our method still largely outperforms the baseline method regarding robust AUC while maintaining a strong anomaly detection performance on clean data.
>
> | Methods|Bird|Plane|Car |Cat| Deer| Dog |Frog| Horse|Ship|Truck| Average|
> | ----| ---- | ---- | ---- | ---- | ---- | ---- | ---- | ---- | ---- | ---- | ----|
> | FastFlow | $0.3^{(63.0)}$ | $4.1^{(74.2)}$ | $1.9^{(81.7)}$ | $0.2^{(45.6)}$ | $0.5^{(56.1)}$ | $1.0^{(72.7)}$ | $0^{(79.7)}$ | $1.2^{(76.4)}$ | $1.8^{(81.2)}$ | $3.7^{(83.7)}$ | $1.5^{(71.4)}$ |
> | CFA | $0.3^{(68.1)}$ | $1.2^{(71.8)}$ | $0.0^{(76.3)}$ | $0.3^{(58.7)}$ | $1.0^{(74.6)}$ | $1.0^{(64.5)}$ | $0.9^{(81.7)}$ | $1.6^{(74.9)}$ | $1.3^{(81.0)}$ | $0.4^{(74.8)}$ | $0.8^{(72.6)}$ |
> | FreeRAD(ours) | $\mathbf{57.9}^{(71.9)}$ | $\mathbf{54.9}^{(79.3)}$ | $\mathbf{50.6}^{(70.4)}$ | $\mathbf{28.7}^{(56.7)}$ | $\mathbf{60.5}^{(71.5)}$ | $\mathbf{39.1}^{(56.4)}$ | $\mathbf{59.6}^{(71.3)}$ | $\mathbf{47.1}^{(60.4)}$ | $\mathbf{65.2}^{(77.5)}$ | $\mathbf{23.2}^{(45.0)}$ | $\mathbf{48.7}^{(66.0)}$ |
>
> | Methods|Bird|Plane|Car |Cat| Deer| Dog |Frog| Horse|Ship|Truck| Average|
> | ----| ---- | ---- | ---- | ---- | ---- | ---- | ---- | ---- | ---- | ---- | ----|
> | FastFlow | $2.1^{(63.0)}$ | $10.3^{(74.2)}$ | $6.9^{(81.7)}$ | $0.8^{(45.6)}$ | $1.6^{(56.1)}$ | $3.7^{(72.7)}$ | $1.5^{(79.7)}$ | $4.6^{(76.4)}$ | $8.0^{(81.2)}$ | $13.1^{(83.7)}$ | $5.3^{(71.4)}$ |
> | CFA | $1.7^{(68.1)}$ | $3.5^{(71.8)}$ | $3.9^{(76.3)}$ | $1.3^{(58.7)}$ | $5.4^{(74.6)}$ | $3.0^{(64.5)}$ | $5.2^{(81.7)}$ | $5.0^{(74.9)}$ | $5.6^{(81.0)}$ | $3.9^{(74.8)}$ | $3.9^{(72.6)}$ |
> | FreeRAD(ours) | $\mathbf{62.2}^{(71.9)}$ | $\mathbf{64.8}^{(79.3)}$ | $\mathbf{61.7}^{(70.4)}$ | $\mathbf{38.9}^{(56.7)}$ | $\mathbf{63.2}^{(71.5)}$ | $\mathbf{45.4}^{(56.4)}$ | $\mathbf{62.5}^{(71.3)}$ | $\mathbf{51.7}^{(60.4)}$ | $\mathbf{68.9}^{(77.5)}$ | $\mathbf{28.5}^{(45.0)}$ | $\mathbf{54.8}^{(66.0)}$ |
>
>
> For the statement on $k$, we agree that our previous statement is specific to low-level anomalies as shown in Figure 2, where anomaly signals are relatively subtle and adversarial perturbations are intrinsically invisible, therefore the noise added from the partial diffusion process could dominate both of them while high-level features of the input are still preserved for reconstruction. When it comes to novelty detection, the added noise will dominate the adversarial perturbations but not the anomaly signal. And the anomaly image simply cannot be well-reconstructed by the diffusion model (as it has not been seen by the model in the training data) and thus may still lead to large reconstruction error.
>
>
> **Q2:** Table 5 and Table 6 only consider 4 classes (Bottle, Grid, Toothbrush, Wood)
>
> **A2:** Please note that we are not cherry-picking the results here. We only report the results for 4 classes in Table 5 and 6 as these experiments are quite time-consuming. Table 5 shows the robustness evaluation results against EOT-PGD attacks which involve 100 steps of gradient estimation process, each of which takes 20 Monte Carlo samples for one single attack. Similarly, Table 6 shows the certified AUC by randomized smoothing, which involves 1000 Monte Carlo sampling for each sample to obtain a certified radius. Hence we perform these experiments on several categories.
>
>
>
> **Q3:** Reference recent reviews here for further reading, e.g. Ruff et al. (2021) or Pang et al. (2021).
>
> **A3:** Thanks for your suggestion. We have added the reference for further reading in the updated version.

---

> ### Author Response · Authors · 2022-11-17
> **A friendly reminder of the rebuttal conclusion**
>
> We thank you for the precious review time and valuable comments. We have provided corresponding responses and additional results, which we believe have addressed your concerns. We are more than happy to further discuss with you on anything that remains unclear or if you have any further suggestions. Thank you!

---

> ### Author Response · Authors · 2022-12-05
> **We sincerely hope the reviewer finds our response helpful.**
>
> Thank you again for your constructive suggestions and insightful comments to help strengthen the paper. We have responded to each of your concerns and provided additional experimental results. We are more than happy to discuss any concerns that you find not fully addressed. If our responses have addressed all your concerns, we would appreciate it if you could update the score. Thank you.

---

### Author Response · Authors · 2022-11-19
**A friendly reminder of the rebuttal conclusion**

Dear reviews,

Thank you again for your thoughtful comments and constructive suggestions. We have provided thorough responses and additional experimental results. As we are approaching the end of the discussion stage, we would appreciate it if you could read our responses and update the scores if your concerns have been addressed. We are glad to further discuss any concerns that you find not fully addressed. Thank you.

Best regards,
Authors

---

### Decision · Program_Chairs · 2023-01-20

**Decision:**

Reject

**Justification For Why Not Higher Score:**

N/A

**Justification For Why Not Lower Score:**

N/A

**Metareview: Summary, Strengths And Weaknesses:**

This paper aims to obtain the adversarial robustness for diffusion model. The presentation is clear, and the idea seems interesting. The reviewers find the experimental evaluation is weak, and the novelty is very limited. It is better to improve the paper based on the comments of the reviewers.